# Gene interaction perturbation network deciphers a high-resolution taxonomy in colorectal cancer

**Zaoqu Liu[1,2,3], Siyuan Weng[1], Qin Dang[4], Hui Xu[1], Yuqing Ren[5], Chunguang Guo[6], Zhe Xing[7], Zhenqiang Sun[4]\*, Xinwei Han[1,2,3]\***

[1]Department of Interventional Radiology, The First Affiliated Hospital of Zhengzhou University, Zhengzhou, China; [2]Interventional Institute of Zhengzhou University, Zhengzhou, China; [3]Interventional Treatment and Clinical Research Center of Henan Province, Zhengzhou, China; [4]Department of Colorectal Surgery, The First Affiliated Hospital of Zhengzhou University, Zhengzhou, China; [5]Department of Respiratory and Critical Care Medicine, The First Affiliated Hospital of Zhengzhou University, Zhengzhou, China; [6]Department of Endovascular Surgery, The First Affiliated Hospital of Zhengzhou University, Zhengzhou, China; [7]Department of Neurosurgery, The Fifth Affiliated Hospital of Zhengzhou University, Zhengzhou, China

**\*For correspondence:**
fccsunzq@zzu.edu.cn (ZS);
fcchanxw@zzu.edu.cn (XH)

**Competing interest:** The authors declare that no competing interests exist.

**Abstract** Molecular subtypes of colorectal cancer (CRC) are currently identified via the snap-shot transcriptional profiles, largely ignoring the dynamic changes of gene expressions. Conversely, biological networks remain relatively stable irrespective of time and condition. Here, we introduce an individual-specific gene interaction perturbation network-based (GIN) approach and identify six GIN subtypes (GINS1-6) with distinguishing features: (i) GINS1 (proliferative, 24%~34%), elevated proliferative activity, high tumor purity, immune-desert, *PIK3CA* mutations, and immunotherapeutic resistance; (ii) GINS2 (stromal-rich, 14%~22%), abundant fibroblasts, immune-suppressed, stem-cell-like, *SMAD4* mutations, unfavorable prognosis, high potential of recurrence and metastasis, immuno-therapeutic resistance, and sensitive to fluorouracil-based chemotherapy; (iii) GINS3 (*KRAS*-inactivated, 13%~20%), high tumor purity, immune-desert, activation of *EGFR* and ephrin receptors, chromosomal instability (CIN), fewer *KRAS* mutations, *SMOC1* methylation, immunotherapeutic resistance, and sensitive to cetuximab and bevacizumab; (iv) GINS4 (mixed, 10%~19%), moderate level of stromal and immune activities, transit-amplifying-like, and *TMEM106A* methylation; (v) GINS5 (immune-activated, 12%~24%), stronger immune activation, plentiful tumor mutation and neoantigen burden, microsatel-lite instability and high CpG island methylator phenotype, *BRAF* mutations, favorable prognosis, and sensitive to immunotherapy and *PARP* inhibitors; (vi) GINS6, (metabolic, 5%~8%), accumulated fatty acids, enterocyte-like, and *BMP* activity. Overall, the novel high-resolution taxonomy derived from an interactome perspective could facilitate more effective management of CRC patients.

## Editor's evaluation

In this paper, the investigators investigate CRC tumor heterogeneity by using a clustering method to understand perturbation of gene networks. The approach is robust and resulted in identification of significant differences in gene expression signals. The investigators identified six distinct gene inter-action networks (GINs) that characterize tumor landscapes with varying degrees of oncogenic driver mutations, immune infiltration, and drug susceptibilities. These results provide a solid contribution to the field that, if validated, could be utilized as a useful predictive and prognostic correlative biomarkers in future clinical trials.

## Introduction

Colorectal cancer (CRC) is a worldwide health issue, representing a heterogeneous and aggressive disease with the leading cause of tumor-associated lethality (*Sung et al., 2021*). Currently, pathological staging is broadly but inadequately used to guide clinical management due to diverse clinical outcomes of patients within the same stage (*Liu et al., 2022*). The inherent heterogeneity between patients hampers the individualized treatment of CRC. Development of molecular classification takes the plunge toward more effective interventions and provides critical insights into CRC heterogeneity (*Guinney et al., 2015*; *Isella et al., 2017*; *De Sousa E Melo et al., 2013*; *Sadanandam et al., 2013*; *Marisa et al., 2013*). However, molecular subtypes with distinctive peculiarities and outcomes are mainly identified based on the snapshot transcriptional profiles, largely ignoring the dynamic changes of gene expressions in a biological system (*Guinney et al., 2015* ; *Isella et al., 2017*; *De Sousa E Melo et al., 2013*; *Sadanandam et al., 2013*; *Marisa et al., 2013*; *Chen et al., 2021a*). Indeed, gene expressions are commonly variable at distinct time points or conditions, so that the subtypes based on expression data are unstable and difficult to reproduce (*Chen et al., 2021b*). Conversely, biological networks remain relatively stable irrespective of time and condition, and could more reliably characterize the biological state of bulk tissues (*Chen et al., 2021a*; *Sahni et al., 2015*; *Li et al., 2019a*). Previous studies have demonstrated that network analysis is well documented and applied in high-dimensional data, performing more robustly and effectively than single-gene approach (*Chen et al., 2021b*; *Sahni et al., 2015*). Nevertheless, most network-based methods merely focus on gene nodes in the biological network, but ignore the interactions among genes.

To tackle this issue, we introduced a rank-based individual-specific gene interaction perturbation approach (*Chen et al., 2021a*), which not only leveraged gene node information but also included vital interaction information in the biological network. Gene interactions are highly conservative in normal samples but broadly perturbed in diseased tissues (*Sahni et al., 2015*). The interaction perturbation within the network can quantify the interaction change for each gene pair. Thus, the overall perturbation of all gene pairs in the background network is reasonably and effectively utilized to characterize the pathological condition at the individual level. Using the individual-specific gene interaction perturbation network-based program, we identified and diversely validated six gene interaction network-based subtypes (GINS1-6) with distinct clinical and molecular peculiarities. Our results provided a high-resolution classification system and improved the understanding of CRC heterogeneity from an interactome perspective.

## Results

### Six CRC subtypes were identified from the gene interaction-perturbation network

To decipher the heterogeneous subtypes from the interaction-perturbation matrix (Materials and methods, *Figure 1*), we selected the representative features that significantly distinguished tumor from normal samples and maintained high variability within all tumor samples for clustering analysis, which formed a network with 1390 genes and 2225 interactions. This new network also met the scale-free distribution ($R=-0.994$, $p<2.2e-16$; *Figure 1—figure supplement 1E*) and was visualized in *Figure 1—figure supplement 1F*.

Consensus clustering analysis (*Wilkerson and Hayes, 2010*) on the discovery cohort with 2,167 CRC samples and 2,225 gene interactions, initially tested potential clustering numbers ($K=2$–$10$). The cumulative distribution function (CDF) curve and the proportion of ambiguous clustering (PAC) score (*Senbabaoğlu et al., 2014*) of the consensus score matrix suggested the optimal $K=6$, which was also achieved from the Nbclust assessment (*Figure 2A and Figure 2—figure supplement 1A-C*). The silhouette statistic was utilized to identify the samples that best represented one of six gene interaction-perturbation network subtypes (GINS), yielding a core set of 1957 CRC samples (*Figure 2—figure supplement 1D*). The Uniform Manifold Approximation and Projection (UMAP)(*Becht et al., 2019*) cast all samples in two-dimensional spatial coordinates, showing good discrimination (*Figure 2B*).

In the six subtypes, age and gender did not differ in distribution (p>0.05; *Figure 2—figure supplement 2A-B*), whereas the clinicopathological stage was more advanced in GINS2 than the other subtypes (p<0.05; *Figure 2—figure supplement 2C-F*). Microsatellite instability (MSI), a well-established biomarker in CRC (*Raskov et al., 2020*), was prominently enriched in GINS5 (p<2.2e-16;

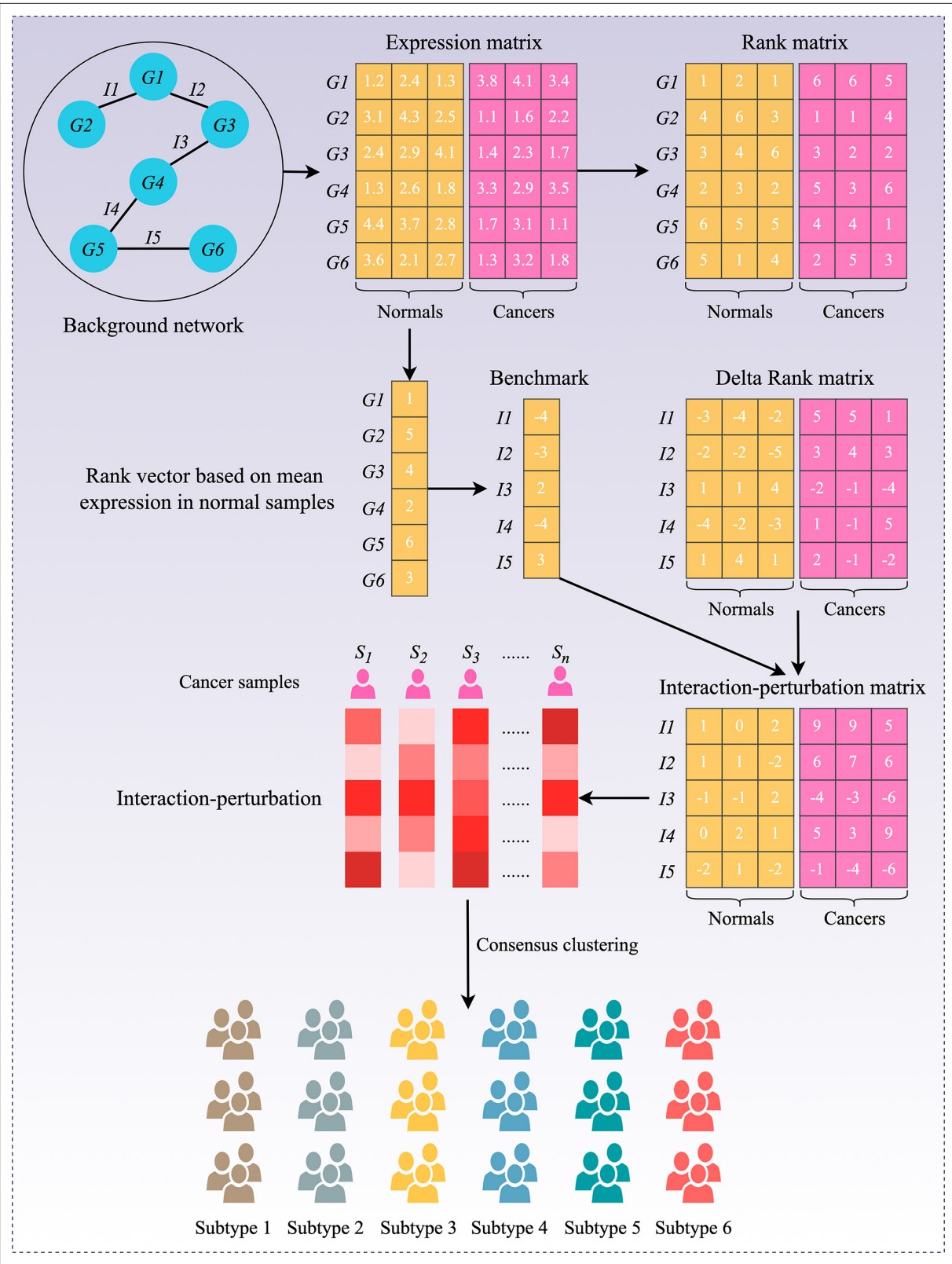

**Figure 1.** Flowchart of the interaction-perturbation-based program. As an example, the background network consists of six genes and five interactions. There were three normal samples (yellow) and three cancer samples (pink). A rank matrix was obtained by ranking the genes according to the expression value of each sample. The rank matrix was converted to a delta rank matrix with five rows and six columns representing interactions and samples,

*Figure 1 continued on next page*

Figure 1 continued

respectively. The benchmark delta rank vector was calculated as the delta rank of the average expression value in all normal samples. The interaction-perturbation matrix was obtained by subtracting the benchmark delta rank vector from the delta rank matrix.

The online version of this article includes the following figure supplement(s) for figure 1:

**Figure supplement 1.** Construction of the gene interaction-perturbation network.

*Figure 2—figure supplement 2G*). Kaplan-Meier survival analysis demonstrated significant survival differences among six subtypes. GINS2 had the worst prognostic outcomes of overall survival (OS) and relapse-free survival (RFS), whereas GINS5 portended the most favorable prognosis, and the other four subtypes displayed intermedium OS and RFS (OS, p<0.0001; RFS, p<0.0001; *Figure 2C–D*). Additionally, GINS2 benefited more from fluorouracil-based adjuvant chemotherapy (ACT) in the discovery cohort with 79 responders and 187 non-responders (p=2.67e-5, *Figure 2E*). We further explored the association of GINS subtypes with ACT after surgery for 585 patients in one subseries of the discovery cohort, GSE39582, which stored complete ACT information. The six subtypes presented concordant distribution in survival across all samples (p=0.0021, *Figure 2—figure supplement 3A*). Subsequent analysis focused on each subtype and revealed that only GINS2 tumors had significantly improved survival after ACT treatment (*Figure 2—figure supplement 3B-G*), suggesting these patients were preferentially responsive to ACT. Conversely, GINS2 might benefit less from bevacizumab in the discovery cohort (25 responders and 29 non-responders), whereas GINS3 possessed a large proportion of responders (p=0.042; *Figure 2F*).

To compare our subtypes with previously reported CRC classifications, the discovery cohort was reclassified according to the previous subtype criteria, including consensus molecular subtypes (CMS) (*Guinney et al., 2015*), CRC intrinsic subtypes (CRIS)(*Isella et al., 2017*), colon cancer subtypes (CCS) (*De Sousa E Melo et al., 2013*), CRCAssigner (CRCA)(*Sadanandam et al., 2013*), and Cartes d'Identité des Tumeurs (*Marisa et al., 2013*), respectively. Noteworthy connections were observed between our subtypes and these previous classifications, indicating a biological convergence (*Figure 2G*). Specifically, GINS1 was related to the canonical CMS2, CCS1, CRCA5, and CIT1; GINS2 was associated with the more aggressive subtypes, including CMS4, CRIS-B, CCS3, CRCA3, and CIT5; GINS3 was linked to CMS2, CRIS-C, CCS1, CRCA2/5, and CIT1; GINS4 was correlated with CMS4, CRIS-A, CCS1/3, and CRCA3; GINS5 was predominantly enriched in MSI-like subtypes, containing CMS1, CRIS-A, CCS2, CRCA4, and CIT2; GINS6 was associated with CMS3, CRIS-A, CCS3, CRCA1, and CIT6. Overall, the aggressiveness properties shown in other classifications were consistent with our six subtypes. Notably, only approximately 50% of our classifier genes overlapped with the signature genes of all previous CRC classifications (*Figure 2—figure supplement 4*), suggesting a significant molecular convergence, but also leaving a rich exploration space for our classification.

## Six subtypes were reproductive and stable in external datasets

To identify GINS subtypes in novel datasets using a small list of genes, a gene centroid classifier was developed. We first identified genes correlated with the six subtypes using significance analysis of microarrays (*Tusher et al., 2001*), followed by prediction analysis for microarrays (*Tibshirani et al., 2002*) to determine 289 subtype-discriminant genes with the lowest misclassification error (1.8%) (*Supplementary file 1*). Subsequently, a 289-gene centroid-based classifier based on the diagonal quadratic discriminant analysis (DQDA) rule (*Marisa et al., 2013*) was developed, and validation datasets were independently assigned to six subtypes. The validation works focused on the following four contexts: (1) data from the same platform (GPL570); (2) data from different platforms and sequencing techniques (microarray or RNA-seq); (3) microdissected or whole tumors; (4) in-house clinical setting. Given the inherent heterogeneity among different datasets, we performed a "correlation of correlations" step as previously reported (*Guinney et al., 2015*).

Initially, significant subtype assignments were performed on seven datasets from the same platform via the 289-gene centroid-based classifier (*Supplementary file 16*). Six subtypes were confidently identified, and Subclass Mapping (SubMap) analysis (*Hoshida et al., 2007*) confirmed that each subtype was associated with similar underlying transcriptional traits in the discovery cohort (*Figure 3—figure supplement 1*). The same results were achieved on seven microarrays from

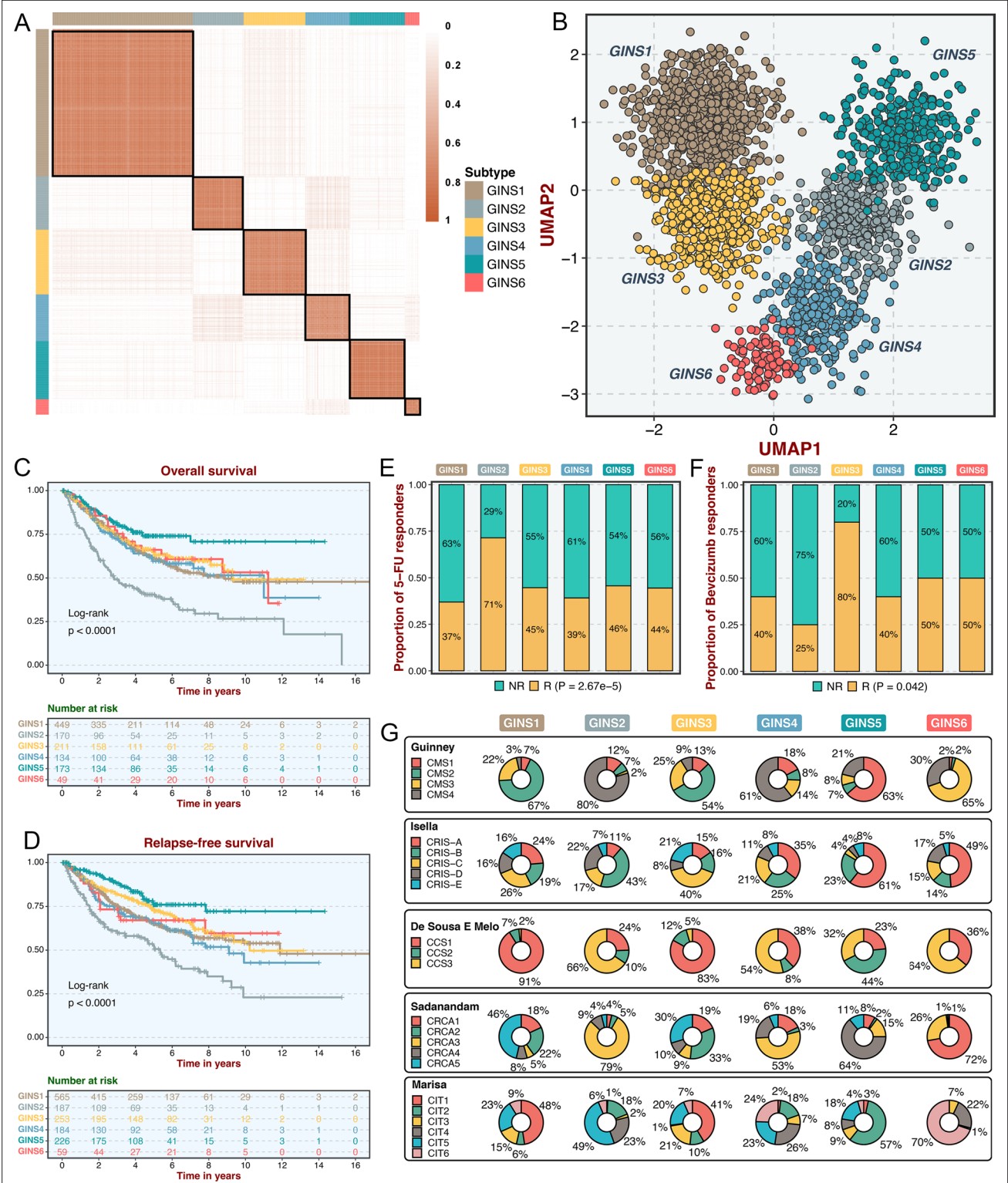

**Figure 2.** Six CRC subtypes were identified from the gene interaction-perturbation network. (**A**) The consensus score matrix of all samples when *K* achieved 6. A higher consensus score between two samples indicates they are more likely to be grouped into the same cluster in different iterations. (**B**) The UMAP analysis cast all samples in two-dimensional spatial coordinates, showing good discrimination. (**C–D**) Kaplan-Meier curves of overall survival and relapse-free survival with log-rank test for six GINS subtypes. Log-rank test. (**E–F**) Barplots showed the distribution of fluorouracil-based adjuvant chemotherapy (**E**) and bevacizumab (**F**) responders in six subtypes. Fisher's exact test. (**G**) Pie charts showed the proportion of other CRC subtypes in the current GINS taxonomy.

*Figure 2 continued on next page*

**Figure supplement 3.** Association of GINS subtypes with ACT after surgery for 585 patients in GSE39582 dataset.

**Figure supplement 4.** The overlapped genes between our classifier genes and the signature genes of all previous CRC classifications.

different platforms and one RNA-seq dataset (TCGA-CRC Illumina; *Figure 3—figure supplement 2*). Two datasets, GSE26682 and GSE24551, each chip from two different platforms (GPL570 & GPL96 for GSE26682 and GPL5175 & GPL11028 for GSE24551), also displayed superimposable classification patterns sustained by similar transcriptional traits (*Figure 3—figure supplement 3*). Proverbially, spatial genetic and phenotypic diversity within solid tumors has been well documented, which is also dubbed as intra-tumor heterogeneity (*Li et al., 2022*). To address this issue, our analysis using additional datasets (GSE12945 and GSE21510) containing samples from both microdissected and whole tumors, and from tumor RNAs profiled on different microarray platforms, consistently reproduced six subtypes with particular molecular traits (*Figure 3A–B*). This is similar to what has been suggested in breast cancer, where subtypes are routinely identified despite possible intra-tumoral heterogeneity (*Polyak, 2011*). In the discovery cohort and 19 validation datasets, we found comparable fractions of patients being assigned to each subtype (*Figure 3C*), which demonstrated that our classification was stable and universal within different datasets. In addition to the identified and attributed subtypes sharing similar transcriptional traits, clinical features were also characterized in validation datasets. Likewise, GINS2 possessed more advanced tumors (*Supplementary file 2*), preferentially metastasized (*Figure 3—figure supplement 4A-F*), and behaved adverse OS (*Figure 3D–H*) and RFS (*Figure 3I* and *Figure 3—figure supplement 4G-H*). The MSI tumors were prone to occur in GINS5 (*Figure 3J–P*) with the most favorable OS (*Figure 3D–H*) and RFS (*Figure 3I* and *Figure 3—figure supplement 4G-H*). ACT treatment also exhibited the identical response distribution, with GINS2 achieving more clinical benefit (*Figure 3Q*). Cetuximab with function to target *EGFR* (*Raskov et al., 2020*), performed better in GINS3 (*Figure 3R*). Overall, six subtypes not only maintained comparable proportions, but also shared analogical transcriptional and clinical traits in the discovery cohort and 19 validation datasets.

## Subtype validation in an in-house clinical cohort

As an initial attempt to facilitate the GINS taxonomy into a clinically translatable tool amenable to clinical applications, we developed a quantitative PCR (qPCR) miniclassifier and further validated our subtypes in 214 clinical CRC samples from our hospital (*Supplementary file 3*). Using 289 genes from the PAM classifier, we firstly identified 93 subtype-specific robust genes via paired differential expression analysis (all p<0.01) and bootstrap logistic regression (1000 iterations and all p<0.05) (*Figure 4A*). Subsequently, the LASSO framework based on 10-fold cross-validation and one-standard-error rule determined the 14 most informative genes that integratively fitted a random forest model (*Figure 4A* and *Supplementary file 4*). Initial model development was conducted in the training dataset (70% of the discovery cohort) and then validated in the testing dataset (30% of the discovery cohort). Confusion matrix displayed the general tendency of classification effect, with a misclassification error of 7.8% and 13.0% in the training and testing datasets, respectively (*Figure 4—figure supplement 1A-B*). The accuracy, precision, recall, F1-score, and specificity of the random forest model reached a quite respectable level, suggesting this miniclassifier comprised of 14 key genes was robust to assign six subtypes in a new cohort (*Figure 4B*).

To test the clinical interpretation of this miniclassifier, another validation based on qPCR results from 214 frozen CRC tissues was deployed to verify its feasibility in clinical settings. With the expression profiles of 14 key genes in each patient, the miniclassifier successfully isolated six subtypes (*Supplementary file 5*). In line with our prior findings, GINS2 had shorter OS and RFS (p<0.0001, *Figure 4C–D*), behaved a stronger propensity to invade and metastasize (p=0.002, *Figure 4E* and *Supplementary file 5*), but was sensitive to ACT treatment (p=0.041, *Figure 4F*). Bevacizumab responders were predominantly concentrated in GINS3, whereas GINS2 still failed to achieve clinical efficacy (*Figure 4G*), although not statistically significant (p=0.390) due to the small sample size

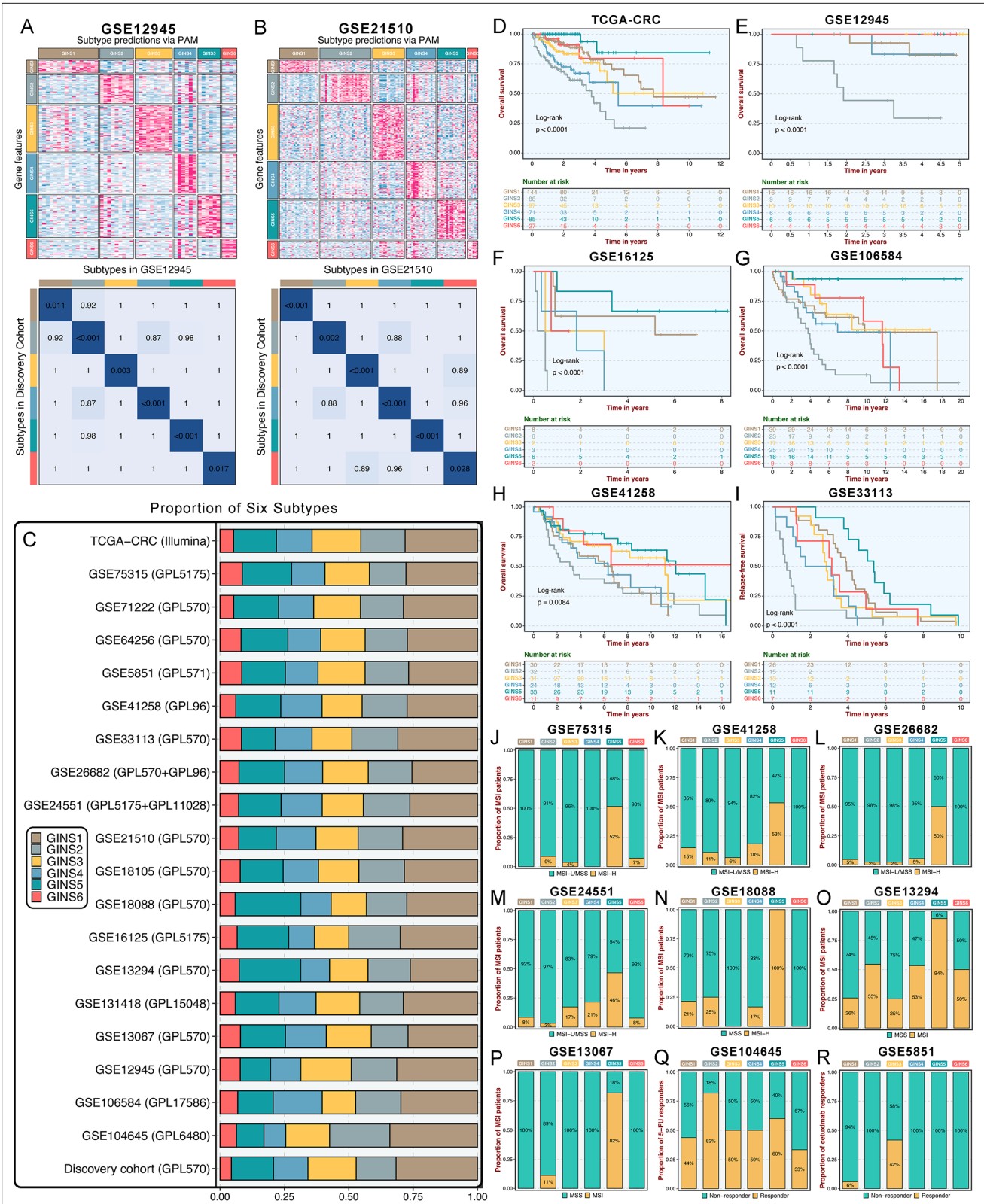

**Figure 3.** Six subtypes were reproductive and stable in external datasets. (**A-B**) The GSE12945 (**A**) and GSE21510 (**B**) were assigned in six subtypes according to the classifier. The top and left bars indicated the subtypes. In the heatmap, rows indicated genes from the classifier and columns represent patients. The heatmap was color-coded on the basis of median-centered $\log_2$ gene expression levels (red, high expression; blue, low expression). SubMap plots, located in the bottom panel, assessed expressive similarity between corresponding subtypes from two different cohorts. (**C**) Barplots

*Figure 3 continued on next page*

*Figure 3 continued*

showed comparable fractions of patients being assigned to each subtype in the discovery cohort and 19 validation datasets. (**D–H**) Kaplan-Meier curves of overall survival for six GINS subtypes in TCGA-CRC (**D**), GSE12945 (**E**), GSE16125 (**F**), GSE106584 (**G**), and GSE41258 (**H**). Log-rank test. (**I**) Kaplan-Meier curves of relapse-free survival for six GINS subtypes in GSE33113. Log-rank test. (**J–P**) Barplots showed the distribution of MSI patients across six subtypes in GSE75315 (**J**), GSE41258 (**K**), GSE26682 (**L**), GSE24551 (**M**), GSE18088 (**N**), GSE13294 (**O**), and GSE13067 (**P**). Fisher's exact test. (**Q–R**) Barplots showed the distribution of responders to six subtypes of fluorouracil-based adjuvant chemotherapy in GSE104645 (**Q**) and cetuximab in GSE5851 (**R**). Fisher's exact test.

The online version of this article includes the following figure supplement(s) for figure 3:

**Figure supplement 1.** Subtype validation of seven datasets from the same platform (GPL570).

**Figure supplement 2.** Subtype validation of seven microarrays from different platforms and a RNA-seq dataset.

**Figure supplement 3.** Subtype validation of two microarrays chip with two different platforms.

**Figure supplement 4.** Clinical characteristics of six GINS subtypes in validation datasets.

(n=42). A subset of CRC, GINS5 (14%), displayed prolonged prognosis (p<0.0001, *Figure 4C–D*) and enriched more MSI tumors (*Figure 4H*). Hence, the 14-gene miniclassifier could afford the stability and interpretation in clinical practice.

## Biological peculiarities of six subtypes

To better delineate the biological attributes inherent to GINS subtypes, we leveraged the 'Hallmark' genesets (*Supplementary file 6*), a comprehensive picture of biological features representing essential oncogenic pathways in cancers (*Hanahan, 2022*). For each sample and pathway, an integrated score was computed by subtracting the average expression of genes negatively correlated with the subtype from the average expression of genes positively correlated with the subtype. To assess the extent to which six subtypes captured samples with stronger transcriptional signatures, we introduced a framework termed 'Sample Set Enrichment Analysis' (SSEA)(*Isella et al., 2017*). In SSEA, all samples are ranked by the integrated scores, and the ranked sample list is further subjected to the gene set enrichment analysis (GSEA) procedure to test whether the 'sample set' for each GINS subtype enriches high-ranking samples. Subsequently, another unsupervised algorithm, gene set variation analysis (GSVA)(*Hänzelmann et al., 2013*), estimated differences in pathway activity across six subtypes.

According to the SSEA and GSVA phenotypic analysis, GINS1 was distinguished by up-regulated cell cycle pathways, suggesting proliferative characteristics for these tumors (*Figure 5A–B*, *Figure 5—figure supplement 1* and *Supplementary file 7*). We next proved that GINS1 also strikingly overexpressed *MKI67* and *PCNA* (p<2.2e-16, *Figure 5—figure supplement 2A-B*), which were identified as important cell cycle-specific antigens in tumors. GINS3 exhibited an inferior level of *KRAS* signaling that was mainly driven by *KRAS* mutations (*Figure 5A–B* and *Supplementary file 7*; *Raskov et al., 2020*). Activation of metabolisms (mainly lipid metabolisms) was featured by GINS6, suggesting canonical metabolic reprogramming across these tumors (*Figure 5A–B*, *Figure 5—figure supplement 1* and *Supplementary file 7*). Intriguingly, interactive stromal and immune activation trends shifted in GINS2/4/5 (*Figure 5A–B*, *Figure 5—figure supplement 1* and *Supplementary file 7*). GINS2 was endowed with higher stromal activity and lower immune activity, whereas GINS5 conveyed the opposite trend entirely, concordant with the tumor invasiveness and prognosis of two subtypes, and GINS4 was characterized by a mixed phenotype that displayed moderate level of stromal and immune pathways. *ESTIMATE* (*Yoshihara et al., 2013*), a tool that uses gene expression profiles to infer immune and stromal constituents within the tumor microenvironment (TME), further validated these phenomena (p<2.2e-16, *Figure 5—figure supplement 2C-D*). As three subtypes with abundant TME components, GINS2/4/5 may mutually evolve in stromal and immune functions. Thus, we intended to extract consistently upregulated and downregulated genes among these three subtypes, using *Mfuzz* package, a noise-robust soft clustering analysis with the fuzzy c-means form (*Kumar and E Futschik, 2007*). The *Mfuzz* analysis revealed 10 gene clusters, and gene cluster 3 and 10 displayed the stable expression pattern from GINS2 to GINS5 (*Figure 5C* and *Supplementary file 8*). As expected, gene cluster 3 was prevailingly associated with immune infiltration and activation (*Figure 5D*), whereas gene cluster 10 was prominently characterized by stromal activation and remodeling (*Figure 5E*), which further supported our findings. This also indicated that TME had

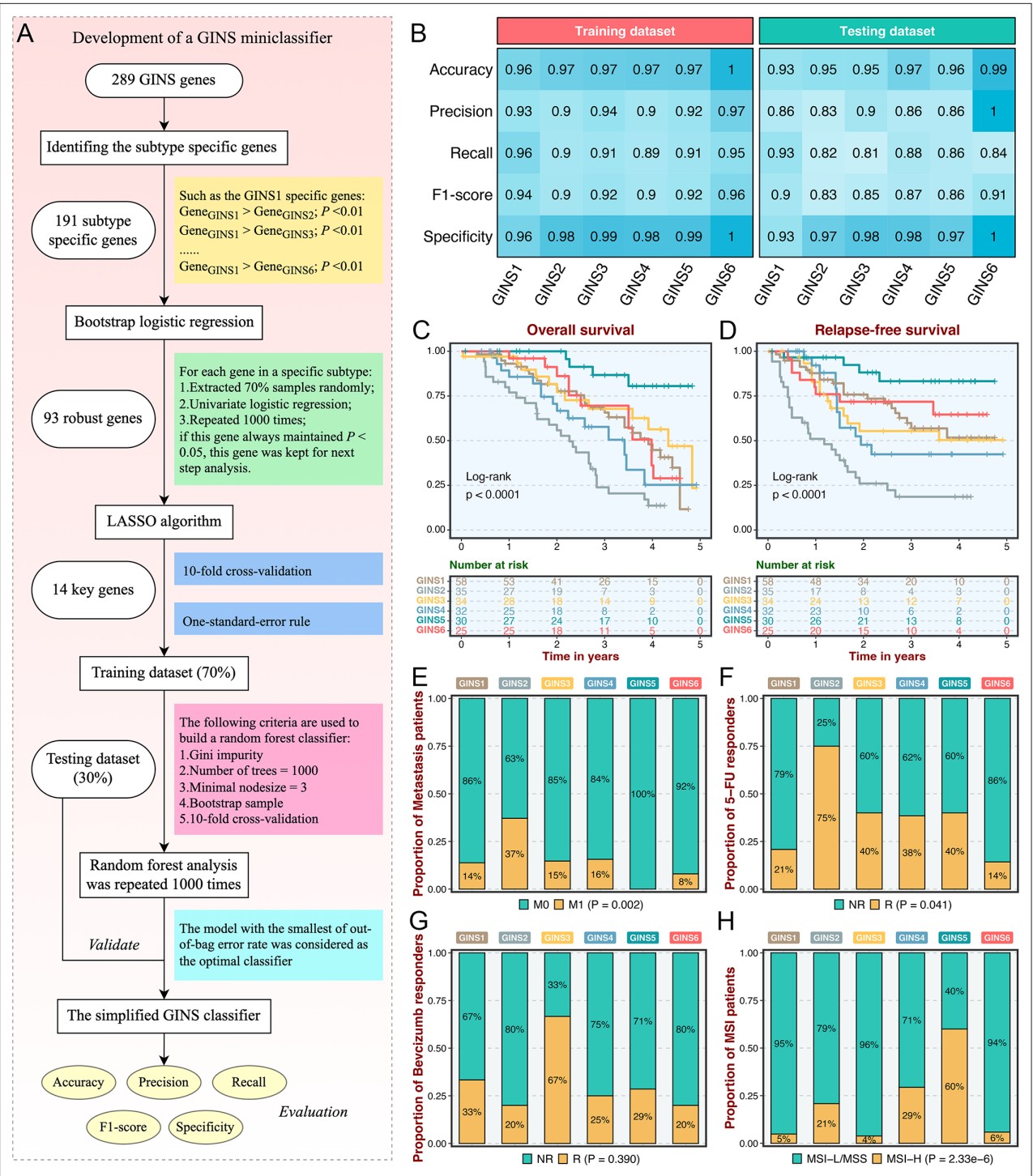

**Figure 4.** Subtype validation in an in-house clinical cohort. (**A**) Overview of the miniclassifier development procedures. (**B**) Performance of the miniclassifier in the training and testing datasets. (**C–D**) Kaplan-Meier curves of overall survival and relapse-free survival with log-rank test for six GINS subtypes. Log-rank test. (**E–H**) Barplots showed the distribution of metastasis patients (**E**), fluorouracil-based adjuvant chemotherapy responders (**F**), bevacizumab responders (**G**), and MSI patients (**H**) in six subtypes. Fisher's exact test.

The online version of this article includes the following figure supplement(s) for figure 4:

**Figure supplement 1.** Performance of the miniclassifier.

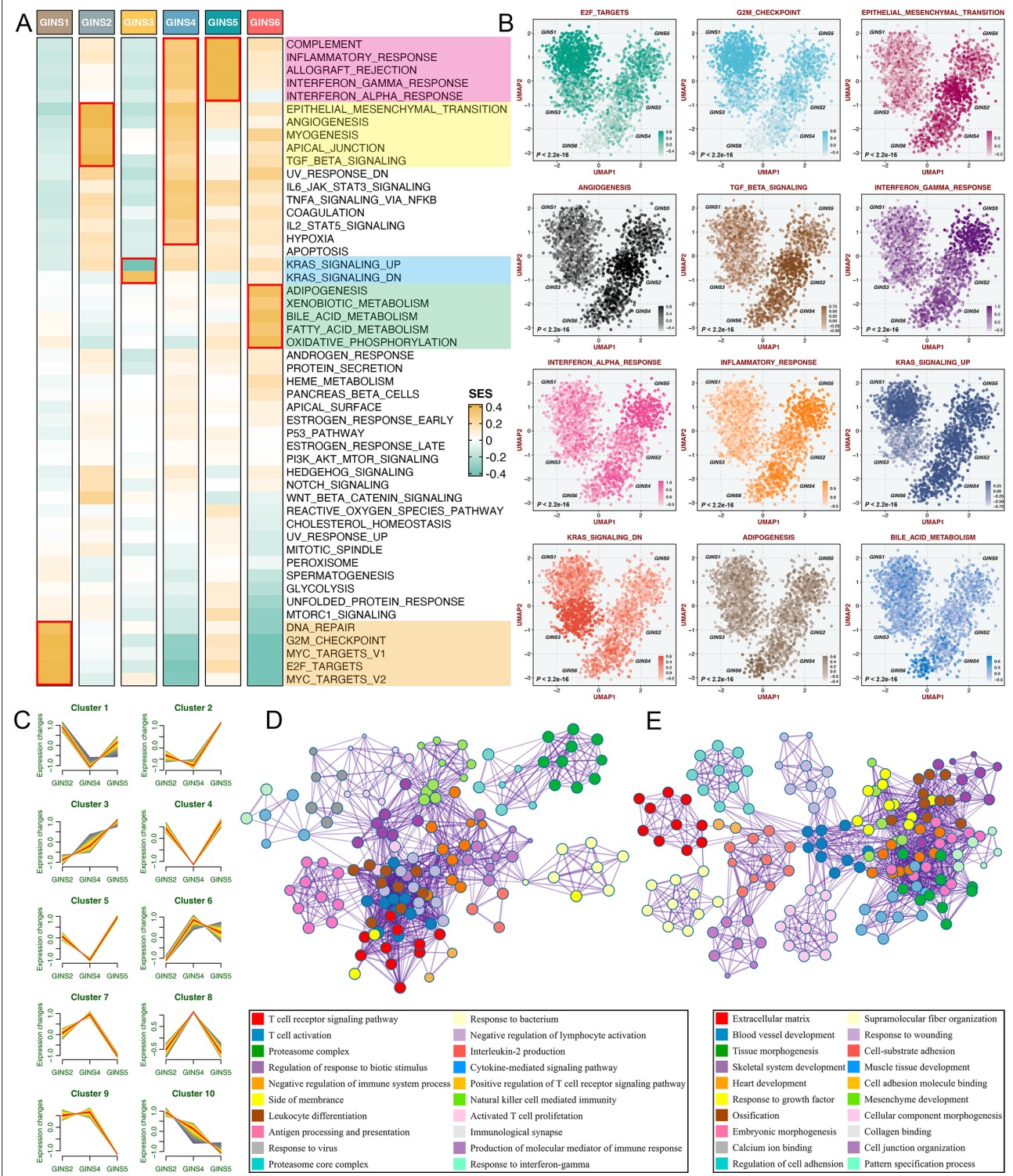

**Figure 5.** Biological peculiarities of six subtypes. (**A**) SSEA-based analysis delineated the biological attributes inherent to GINS subtypes. (**B**) GSVA further estimated differences in pathway activity across six subtypes. Anova test. (**C**) Ten gene clusters were obtained via the soft clustering method (*Mfuzz*) in GINS2/4/5. (**D–E**) Enrichment analysis of gene cluster 3 (**D**) and 10 (**E**).

The online version of this article includes the following figure supplement(s) for figure 5:

**Figure supplement 1.** GSVA estimated differences in pathway activity across six subtypes.

**Figure supplement 2.** The distribution of *MKI67* (**A**), *PCNA* (**B**), stromal score (**C**), immune score (**D**), and tumor purity (**E**) in six subtypes.

profound impacts on the progression and prognosis of tumors, and GINS2/5 acted as two extremes of TME components, indeed showing diametrically opposite clinical outcomes. Of note, GINS1/3 displayed scarce stromal and immune components (*Figure 5A–B*, *Figure 5—figure supplement 1*, and *Figure 5—figure supplement 2C-D*), instead, tumors within these subtypes possessed higher purity (*Figure 5—figure supplement 2E*).

## Immune landscape and immunotherapeutic potential of six subtypes

To further investigate the immune regulations of GINS subfamilies, we profiled five classes of immunomodulators (145 molecules in total), including antigen presentation molecules, immunoinhibitors, immunostimulators, chemokines, and receptors. These immunomodulators are crucial for cancer immunotherapy with specific agonists and antagonists in clinical oncology (*Tang et al., 2018*; *Thorsson et al., 2018*). Our results delineated that transcriptional expression of immunomodulators varied across GINS subtypes, and tumors with high expression pattern of immunomodulators were predominantly assigned to GINS5 (*Figure 6A* and *Supplementary file 9*). To better illustrate this at protein level, we took advantage of the proteome (Reverse Phase Protein Array) data available from the TCGA portal (*Cancer Genome Atlas, 2012*), but with only 26 immunomodulators (*Supplementary file 10*). Using PAM-centroid distance classifier, all samples were attributed to corresponding subtypes. Differential analysis with the thresholds of Benjamini-Hochberg false discovery rate <0.05 and $\log_2$ (fold change)>1 was performed between GINS5 and other subtypes, and we observed 13/26 of immunomodulators were up-regulated in GINS5 (*Figure 6B*). More specifically, 12/13 of significant immunomodulators are involved in antigen presentation, another protein was *IDO1*, an emerging immune checkpoint that overexpresses in multiple cancers (*Zhai et al., 2018*). GINS5 was also characterized by a stronger immunogenicity that harbored remarkably higher tumor mutation burden (TMB) and neoantigen load (NAL) (p<0.001, *Figure 6C*), possibly further inducing abundant immune elements and regulations.

Previous reports introduced several bioinformatics tools based on gene expression profiles to quantify the infiltration and activation of immune cells in solid tumors (*Charoentong et al., 2017*; *Newman et al., 2019*; *Becht et al., 2016*; *Rooney et al., 2015*). Using these tools, we found that rich infiltration and strong immune killing of T cells were particularly evident in GINS5, coincident with the abovementioned findings (*Figure 6D*). Moreover, GINS5 also possessed the abundant infiltration of Th1, Th2, and M1 macrophages (*Mills et al., 2016*; *Figure 6—figure supplement 1A-C*), which could secrete proinflammatory cytokines and enhance immune activation. Conversely, M2 traditionally regarded as promoting tumor growth by suppressing cell-mediated immunity and subsequent cancer cell killing (*Mills et al., 2016*), was significantly elevated in GINS2 (*Figure 6—figure supplement 1D*). In line with this, three other classical immunosuppressive cells, including fibroblasts, myeloid-derived suppressor cells (MDSC), and Treg cells (*Hicks et al., 2022*), were also significantly enriched in GINS2 (*Figure 6E*). Apart from the immune activation represented by GINS5, GINS1/2/3 displayed sparse infiltration of cells that promote immune activity (*Figure 6D* and *Figure 6—figure supplement 1A-C*), but unlike GINS2, GINS1/3 were also characterized by rare immunosuppressive cells (*Figure 6E* and *Figure 6—figure supplement 1D*), consistent with their high tumor purity. GINS4/6 subfamilies were featured as the mixed phenotypes with immune activating and inhibitory components (*Figure 6D–E* and *Figure 6—figure supplement 1A-D*).

To systematically evaluate immunotherapeutic potential of six subtypes, we built an immunogram for the cancer-immunity cycle (CIC) (*Figure 6F*), which was based on the rationale that immunity within tumors is a dynamic process and proposed by Karasaki and colleagues (*Karasaki et al., 2017*). Together, we annotated six subtypes by specific immune features: (i) GINS1/3, thereafter designated the 'immune-desert' phenotype, was endowed with scarce immune fractions; (ii) GINS2, thereafter designated the 'immune-suppressed' phenotype, was enriched for abundant inhibitory cells; (iii) GINS5, thereafter designated the 'immune-activated' phenotype, was dramatically linked to superior tumor immunogenicity and extensive immune activation; and (iv) GINS4/6, thereafter designated the 'mixed' phenotype, was characterized by moderate levels of immunity cycle score (*Figure 6F*).

Among these six subtypes, patients with lower tumor immune dysfunction and exclusion score (*Jiang et al., 2018*), higher immunophenoscore (*Charoentong et al., 2017*) and T-cell-inflamed gene expression profiles (*Ott et al., 2019*), were proven to favor benefit from immunotherapy, and predominantly assigned to GINS5 (*Figure 6G*). SubMap analysis (*Hoshida et al., 2007*) also delineated the

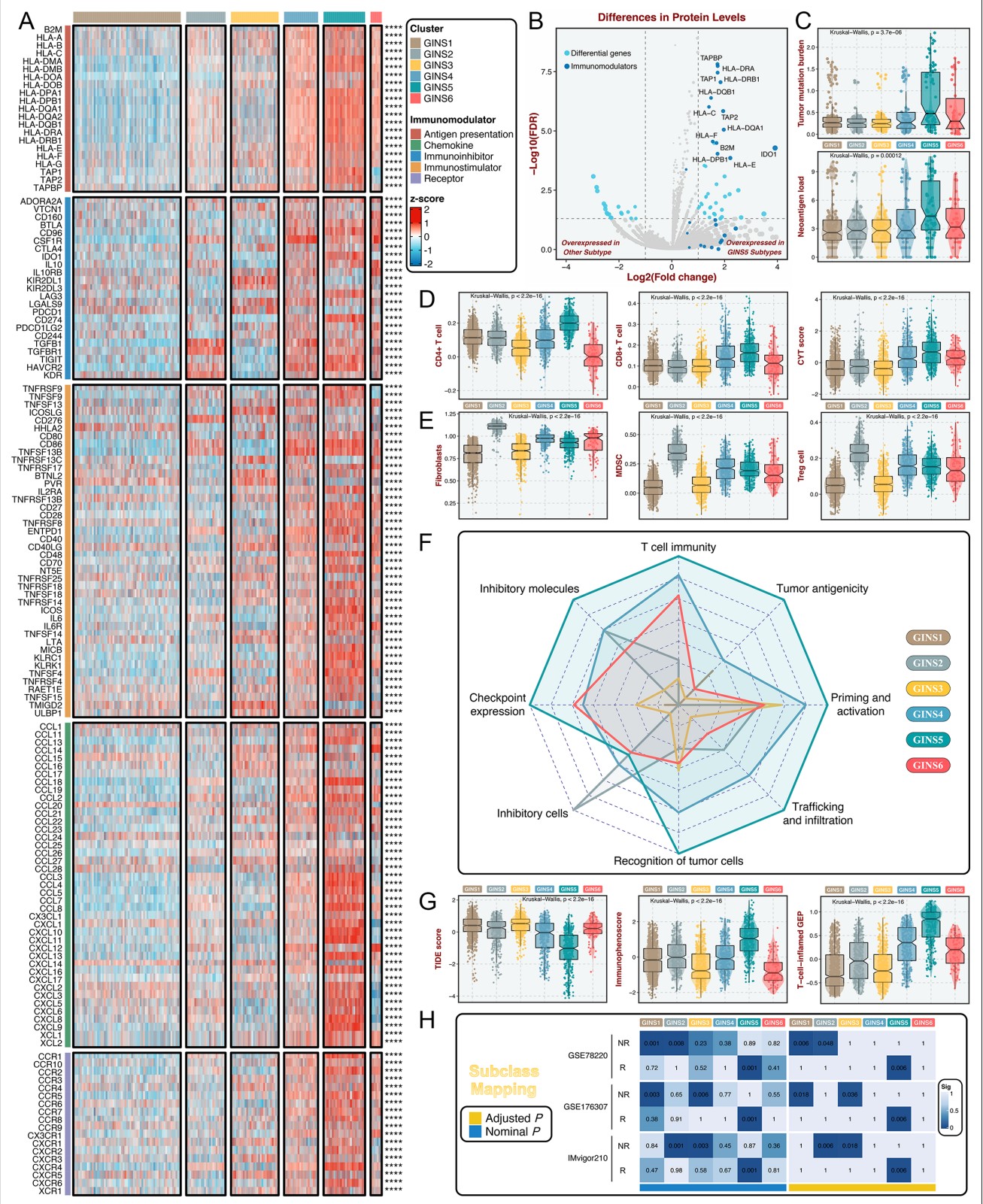

**Figure 6.** Immune landscape and immunotherapeutic potential of six subtypes. (**A**) The expression distribution of 145 immunomodulators among six subtypes. ****p<0.0001. (**B**) Differential analysis was performed between GINS5 and other subtypes, and 13/26 of immunomodulators were up-regulated in GINS5. (**C**) The distribution of TMB and NAL score among six subtypes. Kruskal-Wallis test. (**D**) The distribution of CD4 +T cells, CD8 +T cells, and CYT score among six subtypes. Kruskal-Wallis test. (**E**) The distribution of fibroblasts, MDSC, and Treg cells among six subtypes. Kruskal-Wallis test.

*Figure 6 continued on next page*

*Figure 6 continued*

(**F**) Radar plots showed the immunogram patterns of the six subtypes. Kruskal-Wallis test. (**G**) The distribution of TIDE score, immunophenoscore, and T-cell-inflamed gene expression profiles (GEP) among six subtypes. Kruskal-Wallis test. (**H**) SubMap analysis delineated the similar expression pattern between GISN5 tumors and immunotherapeutic responders from three cohorts with treatment annotations, and GINS1/2/3 shared the transcriptional modes with non-responders from 2/3 of immunotherapeutic cohorts. 'R' represents responder, whereas 'NR' represents non-responder.

The online version of this article includes the following figure supplement(s) for figure 6:

**Figure supplement 1.** Immune cell infiltrations of six subtypes.

similar expression pattern between GISN5 tumors and immunotherapeutic responders from three cohorts with treatment annotations, and GINS1/2/3 shared the transcriptional modes with non-responders from 2/3 of immunotherapeutic cohorts (*Figure 6H*). Collectively, GINS5 tumors might generate clinical benefit from immunotherapy, whereas GINS1/2/3 were not suitable for this treatment due to potential immune-related adverse events and high cost.

## GINS6 tumors conveyed rich lipid metabolisms

Prior results indicated that GINS6 was characterized by activation of metabolism pathways. To investigate an extensive spectrum of metabolic reprogramming in GINS6, we executed GSEA against 69 metabolic pathways from the Kyoto Encyclopedia of Genes and Genomes (KEGG) database (*Chen et al., 2021b*; *Supplementary file 11*). In total, 20 pathways were significantly enriched in GINS6 versus other subtypes, and most pathways were upregulated (*Figure 7A* and *Supplementary file 11*). Notably, GSEA demonstrated that lipid metabolisms were the most significant metabolic processes in GINS6 (*Figure 7A–B* and *Supplementary file 11*). Using principal component analysis (PCA), we found that only the lipid metabolism profiles could distinguish GINS6 from other subtypes in spatial distribution (*Figure 7C* and *Figure 7—figure supplement 1*). The SSEA-based framework further confirmed that GINS6 predominantly enriched high-ranking samples with stronger lipid signature scores (*Figure 7D*). Subsequently, we established a metabolite-protein interaction network (MPIN) (*Chen et al., 2021a*) via nine GINS6-specific genes with broad and tight connections with lipid metabolites (*Figure 7E* and *Supplementary file 12*). Indeed, 7/9 of these genes belonged to lipid metabolic pathways. To explore the metabolic profiles from the perspective of metabolomics, we enrolled 55 CRC cell lines with both transcriptome and metabolomics data (including 225 metabolites) from Cancer Cell Line Encyclopedia (CCLE)(*Li et al., 2019b*). All cell lines were assigned to corresponding subtypes via our PAM-centroid distance classifier. We compared the metabolite abundances between GINS6 and other subtypes, and found that GINS6 exhibited higher levels in four fatty acids including α-glycerophosphate, adipate, taurocholate, and aconitate. Additionally, four carnitines containing stearoylcarnitine, myristoylcarnitine, valerylcarnitine, and malonylcarnitine, that serve as vital compounds in lipid metabolism processes, were also dramatically accumulated in GINS6 (*Figure 7F*). These findings validated that GINS6 was closely associated with metabolic reprogramming and accumulated fatty acids, suggesting GINS6 tumors might be more sensitive to metabolic inhibitors targeting fatty acid metabolisms.

## GINS subtypes were associated with cellular phenotypes and autocrine loops

Using previously supervised signatures derived from cellular phenotypes (*Sadanandam et al., 2013*; *Marisa et al., 2013*; *Kosinski et al., 2007*), we identified phenotype origins peculiar to individual GINS classes. In this study, GINS2 appeared highly enriched for stem-cell-like tumors (91%), whereas GINS4 was endowed with transit-amplifying-like phenotype (86%) (*Figure 8A*). GINS5 was characterized by inflammatory (*Figure 8A*), coincident with its biological and immune features. GINS6 featured an enterocyte-like phenotype (*Figure 8A*). Specifically, serrated-like CRC arising from serrated neoplasia pathway (*Marisa et al., 2013*), were predominantly assigned to GINS5 and to a lesser extent to other subtypes (*Figure 8A*). Conversely, conventional-like tumors were mainly shared by non-GINS5 subtypes. From the unsupervised perspective, GSVA further verified the cellular phenotypic differences across six subtypes (*Figure 8B–H*). As previously reported (*Sadanandam et al., 2013*), we also delineated that these cellular phenotypes were linked to distinctive WNT signaling activity and anatomical regions of the colon crypts (*Figure 8I–J*). Moreover, the nearest template prediction (NTP)

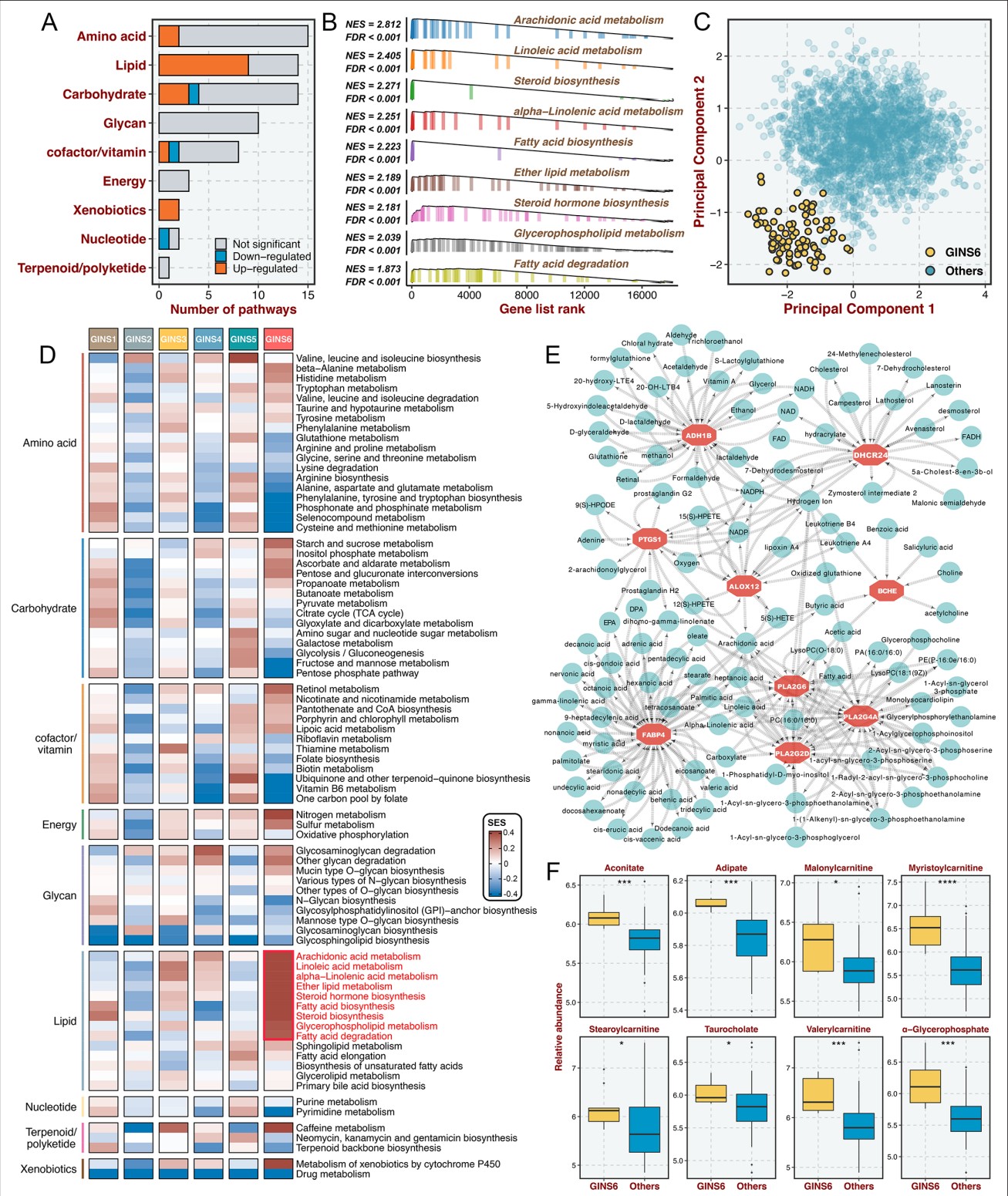

**Figure 7.** GINS6 tumors conveyed rich lipid metabolisms. (**A**) The number of metabolic pathways that was significantly upregulated or downregulated (FDR <0.05) in GINS6 versus the other subtypes among each of nine metabolic categories. (**B**) GSEA plots of nine lipid metabolism pathways (FDR <0.001). (**C**) Principal component analysis of all samples in the discovery cohort for the mRNA expression of lipid metabolic genes. (**D**) SSEA-based framework further confirmed that GINS6 predominantly enriched high-ranking samples with stronger lipid signature scores. (**E**) Metabolite-protein interaction network (MPIN) was established via nine GINS6-specific genes with broad and tight connections with lipid metabolites. (**F**) Metabolomics results further demonstrated that GINS6 featured by abundant fatty acids metabolites. Wilcoxon test. *p<0.05, ***p<0.001, ****p<0.0001.

*Figure 7 continued on next page*

*Figure 7 continued*

The online version of this article includes the following figure supplement(s) for figure 7:

**Figure supplement 1.** Principal component analysis of all samples in the discovery cohort for the mRNA expression of metabolic genes.

algorithm (*Hoshida, 2010*) based on published signatures (*Kosinski et al., 2007*) assigned each sample into the crypt base and top phenotypes in the discovery cohort (*Figure 8K*). Consistently, tumors with the crypt base phenotype were particularly evident for GINS2, whereas other subtypes were mainly concentrated on tumors with the crypt top phenotype, especially GINS6. We next curated 26 published stemness signatures from the StemChecker webserver (*Pinto et al., 2015*) and further employed GSVA to quantify the signature score of each pathway. Overall, GINS2 displayed superior abundance relative to other subtypes, which was in line with its malignant traits (*Figure 8L*).

Using SSEA, we also assessed the biological significance of each subtype in mitogenic/anti-apoptotic autocrine loops (*Isella et al., 2017*), as a proxy of growth factor-dependent oncogenic signaling (*Supplementary file 13*). All samples in SSEA framework were ranked according to 'receptor activation index', which were computed by averaging the expression of receptor and its ligands. As results, GINS1 was mainly associated with elevated *NOTCH2* and *IL13RA2* autocrine stimulation loops (*Figure 8M* and *Figure 8—figure supplement 1*). GINS2 displayed high intrinsic *TGFBR1* and *IGF2R* stimulation (*Figure 8M* and *Figure 8—figure supplement 1*). Of note, GINS3 was characterized by activations of ephrin receptors (*EPHA* and *EPHB* signaling) (*Figure 8M* and *Figure 8—figure supplement 1*), a set of receptors that are activated via binding to Eph receptor interacting proteins and form the largest subfamily of receptor tyrosine kinases (RTKs). In line with prior findings, GINS3 featured a high *EGFR* activity (*Figure 8M* and *Figure 8—figure supplement 1*), corresponding to its sensitive response to cetuximab. GINS4 exhibited marked traits of high activities in *ACKR2*, *FGFR1*, and *IL1R1* stimulation loops (*Figure 8M* and *Figure 8—figure supplement 1*). Accordingly, our results attributed immune-related autocrine loops including *CXCR3*, *IFNAR1*, *IFNGR1*, *TNFRSF9*, and *TNFRSF10A* to GINS5 tumors (*Figure 8M* and *Figure 8—figure supplement 1*), concordant with inflammatory traits of this subtype. GINS6 was linked to *BMP* activity (*Figure 8M* and *Figure 8—figure supplement 1*), which was reported to restrict stem cell expansion and upregulated at the crypt top with a decreasing gradient towards the crypt base (*Kosinski et al., 2007*). Taken together, these findings further provided a higher resolution of GINS taxonomy.

## Multi-omics alteration characteristics of six subtypes

To identify the genetic traits peculiar to individual GINS subfamilies, we characterized the multi-omics landscape in the TCGA-CRC cohort (*Figure 9A*). *PIK3CA* mutations could activate *PI3K/AKT* signaling and further enhance the proliferation and invasion of cancer cells (*Raskov et al., 2020*), which was prevalent in GINS1 (45%) (*Figure 9A*). GINS2 enriched plentiful *SMAD4* mutations (53%), which was strikingly higher than background mutations of *SMAD4* in CRC (*Raskov et al., 2020*; *Figure 9A*). As previously reported, *KRAS* mutations are widespread in CRC (*Raskov et al., 2020*), but to a lesser extent in GINS3 (12%) (*Figure 9A* and *Figure 9—figure supplement 1A*), in line with its inferior activity of *KRAS* signaling detected in the discovery cohort. GINS5 was previously identified as tumors with high TMB and MSI, and thus displayed overall rich mutations in driver genes (*Figure 9A*), especially *BRAF* (*Figure 9—figure supplement 1B*), which was associated with CRC showing a high level of MSI. Conversely, GINS5 presented low chromosomal instability (CIN), featured by slight copy number variation (CNV), whereas an evident CIN phenotype was assigned to GINS3 that possessed heavy CNV burden, including amplifications and deletions (*Figure 9A–D*). We also observed that the broad amplifications of Chr20 were particularly evident for GINS3 (*Figure 9E–F*). *Liu et al., 2019* demonstrated that tumors with TMB-high and CNV-low showed favorable response to immunotherapy, further validating the enhanced remission potential for immunotherapy in GINS5. Subsequently, we identified four CpG island methylator phenotypes (CIMP) from the TCGA-CRC cohort using the beta value of 5,000 CpG island promoters with the most variation (*Figure 9—figure supplement 1C*). As previously reported, high CIMP (CIMP-H) was parallel with high MSI (MSI-H)(*Raskov et al., 2020*), and our results consistently displayed that tumors with MSI-H or CIMP were predominantly assigned to GINS5 (*Figure 9G*). In this study, we determined seven DNA methylation-driven genes via our introduced pipeline (*Figure 9A*). Specifically, the methylation silencing of *SMOC1* was strongly enriched in GINS3, and *TMEM106A* silencing prevalently occurred in GINS4 (*Figure 9A*). The expression levels

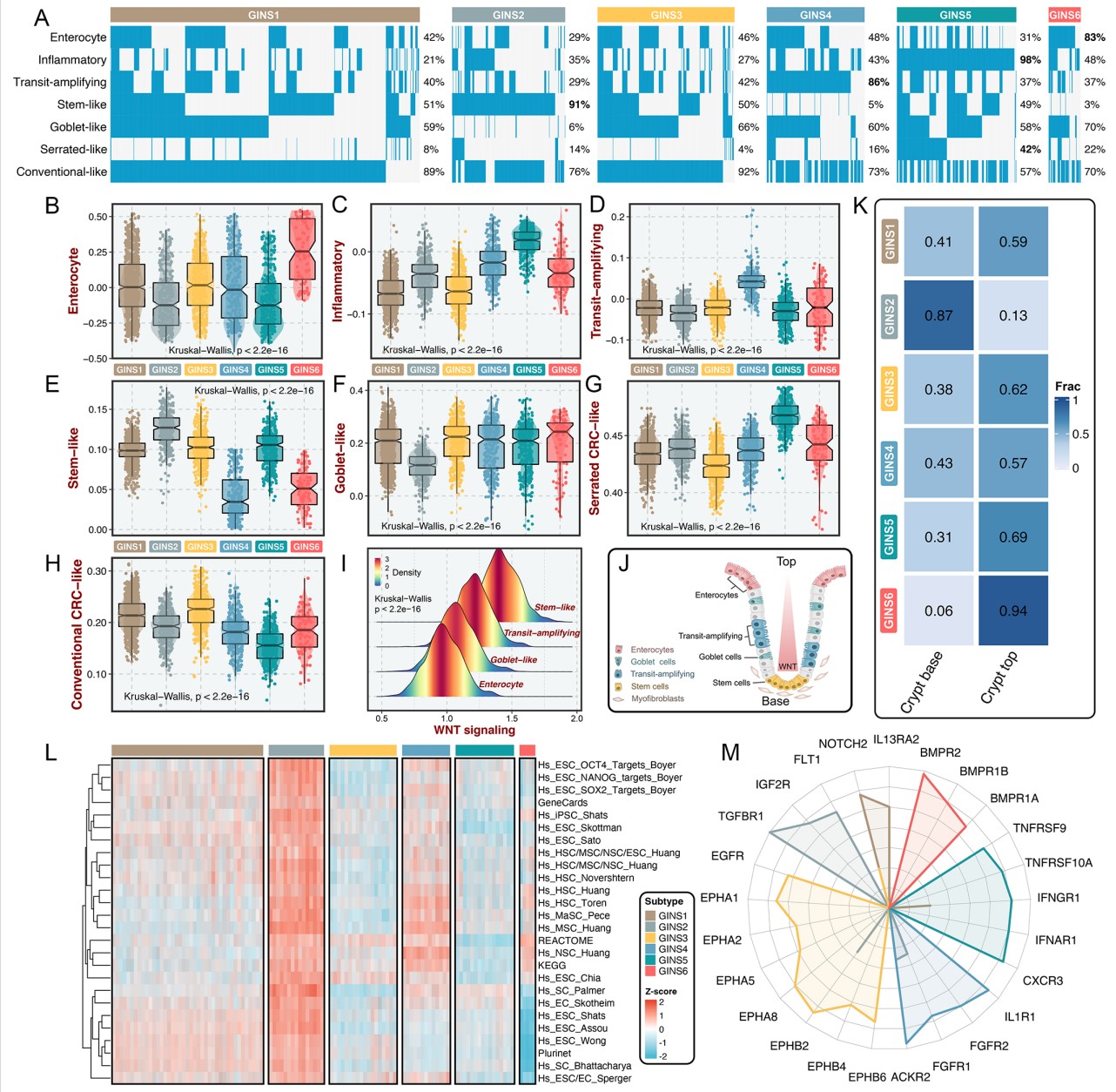

**Figure 8.** GINS subtypes were associated with cellular phenotypes and autocrine loops. (**A**) Supervised approach identified phenotype origins peculiar to individual GINS classes. (**B–H**) Unsupervised-based GSVA showed the distribution of enterocyte (**B**), inflammatory (**C**), transit-amplifying (**D**), stem-like (**E**), goblet-like (**F**), serrated CRC-like (**G**), conventional CRC-like (**H**) scores among six subtypes. Kruskal-Wallis test. (**I**) The distribution of WNT signaling score in different cell-like tumors. Kruskal-Wallis test. (**J**) CRC cellular phenotypes correlated with colon-crypt location and WNT signaling. (**K**) Fractions of the crypt base and top phenotypes among six subtypes. Nearest template prediction (NTP) algorithm based on published signatures assigned each sample into the crypt base and top phenotypes in the discovery cohort. (**L**) GSVA analysis revealed that GINS2 displayed superior stemness abundance relative to other subtypes. (**M**) Radar plot showed autocrine stimulation loops in GINS subtypes.

The online version of this article includes the following figure supplement(s) for figure 8:

**Figure supplement 1.** GSVA further estimated differences in Receptor-Ligand activities across six subtypes.

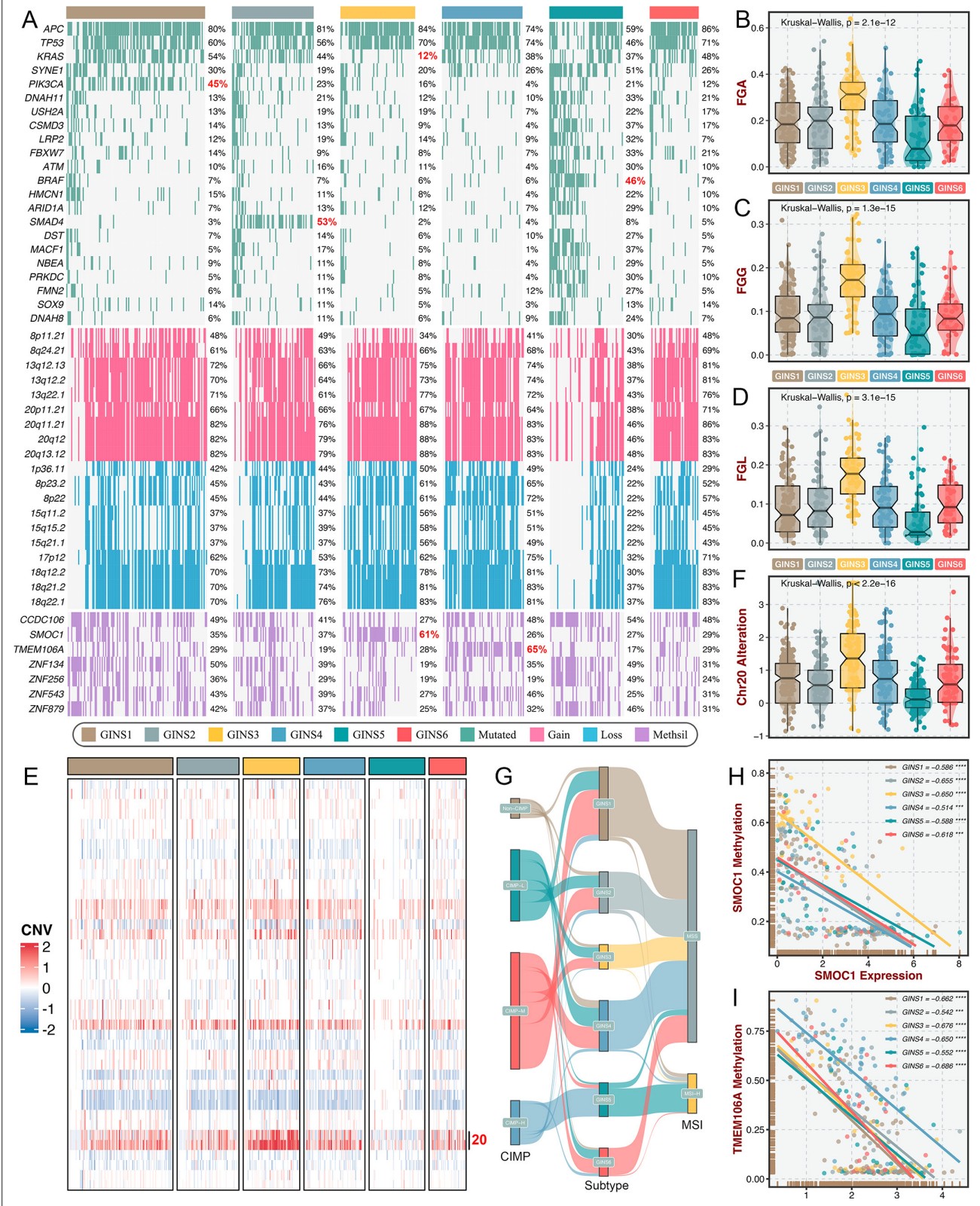

**Figure 9.** Multi-omics alteration characteristics of six subtypes. (**A**) Genomic alteration landscape according to GINS taxonomy. The mutational genes were selected based on mutation frequency >10% and MutSigCV q-value <0.05. The focal gain and loss regions were selected based on CNV frequency >40% and GISTIC q-value <0.05. The methylation silencing (Methsil) genes were identified based on our pipeline. (**B–D**) The distribution of fraction of genome alteration (FGA), fraction of genome gained (FGG), and fraction of genome lost (FGL) among six subtypes. Kruskal-Wallis test.

*Figure 9 continued on next page*

Figure 9 continued

(**E**) Heatmap showed the distribution of broad copy number changes across six subtypes in the TCGA-CRC dataset. (**F**) The distribution of Ch20 alterations in six subtypes. Kruskal-Wallis test. (**G**) Sankey plot showed connections among GINS subtypes, CIMP phenotypes, and MSI phenotypes. (**H–I**) The expression levels of *SMOC1* and *TMEM106A* were significantly inversely correlated with their methylation levels. ***p<0.001, ****p<0.0001.

The online version of this article includes the following figure supplement(s) for figure 9:

**Figure supplement 1.** Genomic alterations of six subtypes.

of these two genes were significantly inversely correlated with their methylation levels (*Figure 9H–I*). Collectively, these findings suggested that GINS subtypes were endowed with specific genetic alterations that presumably drive biological characteristics.

## Stromal contribution to the subtype transitions

The tumor transcriptome originated from cancer cells and TME, thus, it is conceivable that stromal components might impact the subtype assignments of CRC. Previous reports suggested that the subtype derived from stromal contents is a strong indicator of tumor aggressiveness and poor prognosis (*Isella et al., 2017*; *Isella et al., 2015*), which was consistent with the inherent characteristics of stromal-derived GINS2 subtype. Indeed, most of GINS2-discriminant genes from the PAM classifier belonged to stromal genes (71.1%), followed by GINS4 (47.5%) (*Figure 10—figure supplement 1A*). To explore the extent of stromal contribution to the GINS subclasses, we leveraged the transcriptional profiles from CRC patient-derived xenografts (PDXs), for which the transcriptome is a mixture of human RNAs (deriving from cancer cells) and mouse RNAs (deriving from stromal cells) (*Figure 10A*). Hence, the stromal transcriptome of PDX samples can't be detected by human microarray or RNA-seq[44]. In Uronis cohort (chip data)(*Uronis et al., 2012*) with 27 matched human CRC samples and PDX derivatives, the subtype assignments were incongruent between PDXs and their original counterparts. Subtypes with rich stromal components (e.g. GINS2 and GINS4) in human CRC samples were inclined to transform into subtypes with high tumor purity (e.g. GINS1 and GINS3) in corresponding PDX derivatives (*Figure 10B*). Another RNA-seq cohort with larger samples, Isella cohort (*Isella et al., 2017*), including 116 matched liver metastatic CRC and mouse xenografts, further validated these findings (*Figure 10C*).

Furthermore, a dataset (GSE56699) comprised 11 pairs of preoperative radiotherapy specimens and matched post-treatment surgical specimens (*Isella et al., 2015*), was utilized to investigate how the substitution of cancer cells by fibrous tissue, a typical reparative reaction triggered by radiotherapy, impacted the subtype assignments. We observed that the pretreatment specimens were confidently assigned to six subtypes, and most of the matched post-treatment biopsies were assigned to GINS2 (*Figure 10D*), confirming that stromal component served as the driven factor for GINS2 transitions. Additionally, in a single cell RNA-seq cohort derived from 11 patients with CRC (*Li et al., 2017*), UMAP projected all cells annotated by reference component analysis (RCA) in spatial distribution (*Figure 10E*). We applied the PAM-centroid distance classifier to perform subtype assignments for these 11 samples (*Figure 10—figure supplement 1B-C*). GINS2 displayed a strikingly higher fraction of fibroblasts relative to other subtypes (*Figure 10—figure supplement 1D*). A previous study reported that tumors with a high level of fibroblasts were resistant to radiotherapy (*Isella et al., 2015*). Thus, we further examined the associations between the GINS subtypes and radiotherapy in GSE56699. As expected, GINS2 possessed superior cancer-associated fibroblasts (CAF) score and worse prognosis across all treated samples (*Figure 10—figure supplement 2A-B*). *Isella et al., 2015* demonstrated that CAF score was a stronger indicator of negative prognosis. In GINS2 samples, a higher CAF score certainly predicted worse prognosis (*Figure 10—figure supplement 2*). We also observed that resistant tumors were predominantly enriched in GINS2 (*Figure 10—figure supplement 2D*). In summary, stromal signals remarkably contributed to the transitions of GINS2, which featured rich fibrous components and was resistant to radiotherapy.

## Identification of potential therapeutic agents for six subtypes

To facilitate the subtype-based targeted interventions, we introduced an integrated pipeline to identify potential therapeutic agents for each subtype (*Yang et al., 2021a*; *Figure 10—figure supplement 3A*). Three pharmacogenomic datasets, CTRP, PRISM, and GDSC, store large-scale drug response and

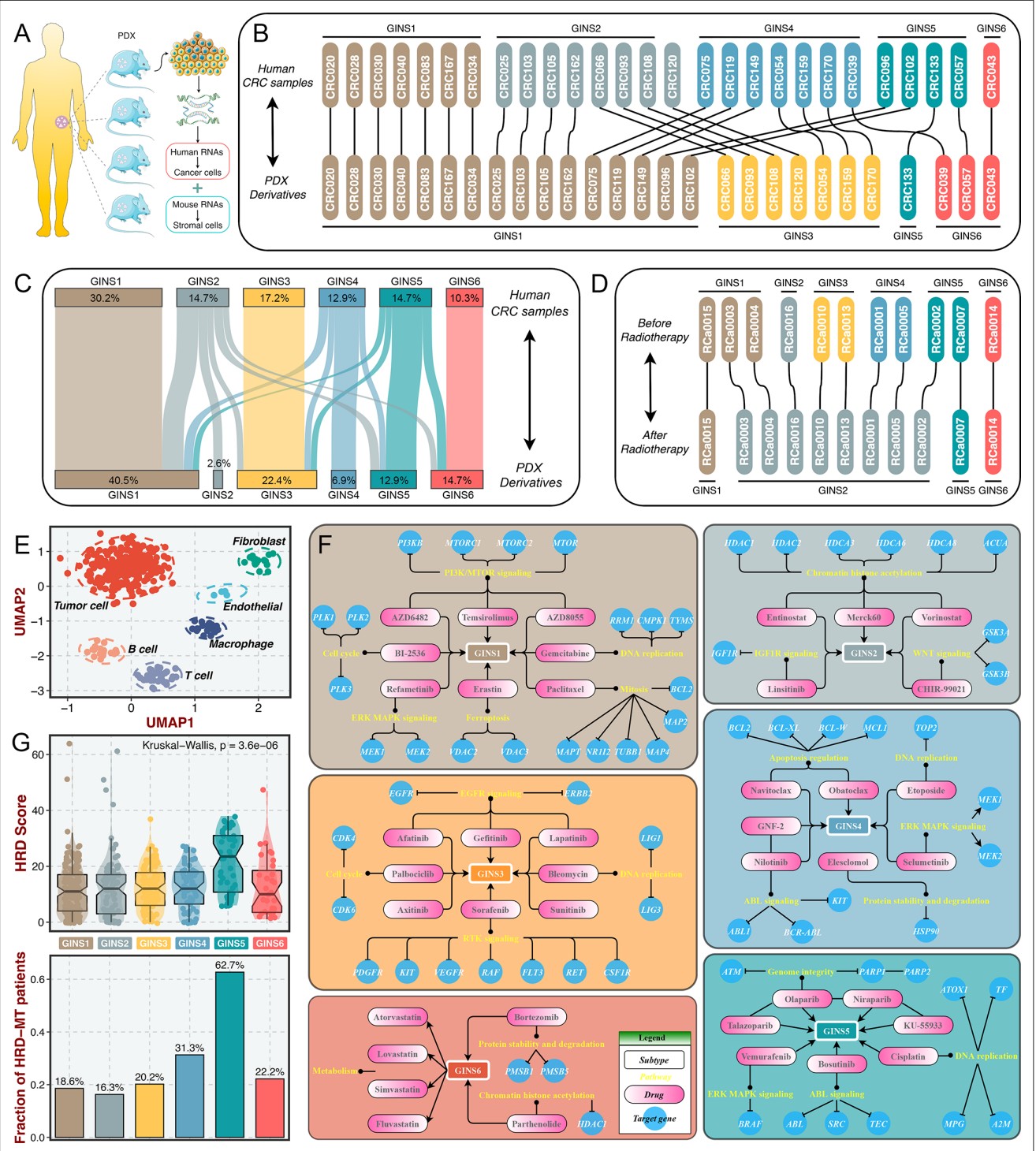

**Figure 10.** Stromal contributions and potential therapeutic agents. (**A**) Schematic diagram showed the PDX transcriptome is a mixture of human RNAs (deriving from cancer cells) and mouse RNAs (deriving from stromal cells). (**B–C**) Transcriptional classification of paired human CRC samples and PDX derivatives in Uronis cohort (**B**) and Isella cohort (**C**). (**D**) Transcriptional classification of paired samples before and after radiotherapy. (**E**) UMAP projected all cells annotated by reference component analysis (RCA) in spatial distribution. (**F**) Identification of potential therapeutic agents for six subtypes. (**G**) The distribution of HRD score and HRD pathway mutations among six subtypes. Kruskal-Wallis test.

The online version of this article includes the following figure supplement(s) for figure 10:

**Figure supplement 1.** Stromal contribution to the subtype transitions.

**Figure supplement 2.** Clinical significance of six subtypes in GSE56699.

*Figure 10 continued on next page*

*Figure 10 continued*

**Figure supplement 3.** Identification of potential therapeutic agents for six subtypes.

**Figure supplement 4.** Candidate agents of six subtypes.

molecular data of human cancer cell lines, enabling accurate prediction of drug response in clinical samples (*Yang et al., 2021b*). As mentioned above, stromal components could obscure the expression patterns of cancer cells in clinical samples. A purification algorithm termed *ISOpure* (*Quon et al., 2013*) was adopted to eliminate the contamination of stromal signal in the discovery cohort prior to conducting drug response prediction, and further yielded a purified tumor expression profiles comparable to cell lines (*Yang et al., 2021a*). After purification, the proportion of stromal-rich subtypes (e.g., GINS2 and GINS4) was obviously decreased, suggesting the impact of stroma components had been eliminated (*Figure 10—figure supplement 3B*). A PDX dataset, GSE73255 (*Isella et al., 2017*), is naturally uncontaminated by human stromal components. Hence, we tested our pipeline in the discovery cohort and GSE73255, and ultimately identified intersecting subtype-specific agents in two datasets (*Figure 10—figure supplement 3A*). To demonstrate the stability of drug response assessment, we examined whether the estimated response of four *EGFR* pathway inhibitors was concordant with their clinical efficacy—with a stronger clinical benefit in *KRAS*-mutant patients (*Raskov et al., 2020*). Our results indicated that patients with *KRAS* mutations squinted towards possessing a lower drug response (*Figure 10—figure supplement 3C*), in line with how *EGFR* pathway inhibitors behaved clinically (*Raskov et al., 2020*).

Taking the intersections of two datasets, we determined 41 specific-subtype agents for six subtypes (*Figure 10—figure supplement 4* and *Supplementary file 14*). Interestingly, the targeted pathways of several candidate drugs were consistent with the biological and genomic peculiarities of corresponding subtypes (*Figure 10F*). For example, GINS1-specific drugs, BI-2536, gemcitabine, and paclitaxel target proliferative pathways; AZD6482, AZD8055 and temsirolimus target activated *PI3K/mTOR* signaling arise from *PIK3CA* mutations, which was strikingly harbored in GINS1; GINS2-specific drugs, linsitinib targets *IGF1R* signaling and CHIR-99021 targets WNT signaling; six *EGFR/RTK* signaling inhibitors, afatinib, gefitinib, lapatinib, axitinib, sorafenib, and sunitinib were specifically designed for GINS3; GINS6 featured dysregulated lipid synthesis, which might be targeted by four anticholesterol drugs containing atorvastatin, lovastatin, simvastatin, and fluvastatin. These results not only identified candidate compounds for each subtype, but also supported our previous findings. Notably, we observed that four *PARP* inhibitors were specific for GINS5, including olaparib, niraparib, talazoparib, and KU-55933 (*Figure 10F*). Previous reports have demonstrated that tumors with homologous recombination deficiency (HRD) are sensitive to *PARP* inhibitors (*Liu et al., 2021a*). In the TCGA-CRC cohort, we next compared the HRD score and the proportion of HRD pathway mutations (*Liu et al., 2021b*) among six subtypes. As expected, tumors with higher HDR score and mutations were predominantly assigned to GINS5 (*Figure 10G*), suggesting its stronger potential to benefit from *PARP* inhibitors. Overall, we provided more subtype-based targeted interventions for GINS taxonomy.

## Discussion

To address the snapshot effect of transcriptional analysis (*Chen et al., 2021b*; *Sahni et al., 2015*; *Li et al., 2019a*), we leveraged a relatively stable gene interaction network to discover the heterogeneous subtypes of CRC from an interactome perspective. As previously reported, biological networks maintain relatively constant irrespective of time and condition, preferably characterizing the biological state of bulk tissues (*Chen et al., 2021a*; *Sahni et al., 2015*; *Li et al., 2019b*). In the biological network, gene interactions are highly conservative in normal samples but broadly perturbed in diseased tissues (*Sahni et al., 2015*). Here, we constructed a large-scale interaction perturbation network from 2,167 CRC tissues and 308 normal tissues, deciphering six GINS subtypes with particular clinical and molecular peculiarities. Notably, although the GINS subtypes were dramatically associated with published classifications, only a limited overlap between our classifier genes with the signature genes of all previous classifications, suggesting a significant molecular convergence but also distinct specialties.

Considering that the stability and reproducibility of molecular subtypes are fundamental for clinical application, the GINS taxonomy was rigorously validated in 19 external datasets (n=3420) with distinct

| | GINS1 | GINS2 | GINS3 | GINS4 | GINS5 | GINS6 |
|---|---|---|---|---|---|---|
| Proportion | 24% ~ 34% | 14% ~ 22% | 13% ~ 20% | 10% ~ 19% | 12% ~ 24% | 5% ~ 8% |
| Tumor purity | High | Low | High | Low | Low | Low |
| Biological peculiarities | Proliferative | Stromal-rich | KRAS inactivation | Mixed | Immune activated | Lipid metabolism |
| Tumor antigenicity | + | - | - | ++ | ++++ | + |
| Antigen presentation | - | - | + | ++ | ++++ | + |
| Trafficking and infiltration | - | + | - | ++ | ++++ | - |
| Priming and activation | - | ++ | ++ | +++ | ++++ | ++ |
| T cell immunity | - | + | - | +++ | ++++ | ++ |
| Inhibitory cells | - | ++++ | - | ++ | + | + |
| Inhibitory molecules | - | +++ | - | +++ | ++++ | ++ |
| Checkpoint expression | - | - | + | +++ | ++++ | +++ |
| Cellular phenotypes | | Stem-cell-like Crypt base | | Transit-amplifying | Inflammatory Serrated-like CRC | Enterocyte |
| Autocrine stimulation loops | NOTCH2 IL13RA2 | TGFBR1 IGF2R | EGFR EPH receptors | AXKR2; FGFR1 IL1R1 | CXCR3; IFNAR1 IFNGR1; TNFRSF9; TNFRSF10A | BMP receptors |
| Genomic phenotypes | | | CIN | | TMB-high; MSI CIMP-H; CIN-L | |
| Multi-omics alterations | PIK3CA-mut | SMAD4-mut | Fewer KRAS-mut Chr20-gain SMOC1-met | TMEM106A-met | BRAF-mut | |
| Prognosis | Intermediate | Worst | Intermediate | Intermediate | Best | Intermediate |
| Metastasis tendency | Intermediate | High | Intermediate | Intermediate | Low | Intermediate |
| Fluorouracil-based ACT | | Sensitive | | | | |
| Cetuximab | | | Sensitive | | | |
| Bevacizumab | | | Sensitive | | | |
| Immunotherapy | Resistant | Resistant | Resistant | | Sensitive | |
| Radiotherapy | | Resistant | | | | |
| Candidate agents | AZD6482 AZD8055 Temsirolimus BI-2536 Gemcitabine Refametinb Erastin Paclitaxel | Entinostat Merck60 Vorinostat Linsitinib CHIR-99021 | Afatinib Gefitinib Lapatinib Palbociclib Axitinib Sorafenib Bleomycin Sunitinib | Navitoclax Obatoclax Etoposide GNF-2 Nilotinib Elesclomol Selumetinib | Olaparib Niraparib Talazoparib KU-55933 Vemurafenib Bosutinib Cisplatin | Atorvastatin Lovastatin Simvastatin Fluvastatin Bortezomib Parthenolide |

**Figure 11.** Summary of the main characteristics of six GINS subtypes.

conditions. Our six subtypes not only maintained comparable proportions, but also shared analogical transcriptional and clinical traits in the discovery cohort and 19 validation datasets. To provide a rapidly accessible clinical tool, we shrunk the 289-gene centroid-based classifier into a 14-gene random-forest miniclassifier. The qPCR results from 214 clinical CRC samples further demonstrated the 14-gene miniclassifier could afford the stability and interpretation in clinical practice.

Importantly, the GINS taxonomy also conveyed clear biological and molecular interpretability and laid a foundation for future clinical stratification and subtype-based targeted interventions (*Figure 11*).

GINS1, a proliferative subtype (24%~34%), is endowed with elevated proliferative activity, high tumor purity, immune-desert, and *PIK3CA* mutations. This subtype displays a moderate malignant phenotype in spite of the high tumor purity, coincident with a previous study (*Mao et al., 2018*). In addition, GINS1 tumors reasonably develop resistance to immunotherapy due to their lower TMB and immune-desert TME. Indeed, current findings didn't reveal an effective intervention for GINS1 in clinical settings. To further improve clinical outcomes of this subtype, we identified eight potential therapeutic agents for GINS1, including BI-2536/gemcitabine/paclitaxel targeting proliferative pathways, AZD6482/AZD8055/temsirolimus targeting activated *PI3K*/*mTOR* signaling arise from *PIK3CA* mutations, refametinib, and erastin.

GINS2, a stromal-rich subtype, is characterized by abundant fibrous content, immune-suppressed, stem-cell-like, *SMAD4* mutations, unfavorable prognosis, and high potential of recurrence and metastasis. In line with previous studies (*Guinney et al., 2015*; *Isella et al., 2017*; *De Sousa E Melo et al., 2013*; *Sadanandam et al., 2013*; *Marisa et al., 2013*; *Isella et al., 2015*), CRC patients with stem and mesenchymal transcriptional traits squint towards displaying the malignant phenotypes. Notably, stromal contents remarkably contributed to the transitions of GINS2 into other subtypes, suggesting PDX or cell lines are not applicable surrogates for assessing the GINS taxonomy (*Sadanandam et al., 2013*). For GINS2 tumors, patients are suitable for fluorouracil-based ACT but resistant to radiotherapy due to a high level of fibroblasts (*Isella et al., 2015*). Unlike GINS1/3, the immunotherapeutic resistance of GINS2 is mainly due to highly infiltrating immunosuppressive cells, such as fibroblasts, MDSC, Treg cells, and M2 macrophages, and is therefore dubbed as the immune-suppressed phenotype.

GINS3, a *KRAS*-inactivated subtype, was featured by high tumor purity, immune-desert, activation of *EGFR* and ephrin receptors, CIN, fewer *KRAS* mutations, and *SMOC1* methylation. This subtype is a typical wild-type *KRAS* subgroup, paralleling by the sensitivity to *EGFR*/*VEGFR* inhibitors, such as cetuximab and bevacizumab. Moreover, we also identified six *EGFR*/*RTK* signaling inhibitors, including afatinib, gefitinib, lapatinib, axitinib, sorafenib, and sunitinib, which could serve as additional supplements for routine agents. Different from GINS1/2, the immunotherapeutic resistance of GINS2 could be driven by sparse immune storage and high burden of CNV (*Liu et al., 2019*). As previously reported, CNV-high tumors tend to respond unfavorably to immunotherapy (*Liu et al., 2019*).

GINS4, a mixed subtype, is distinguished by moderate level of stromal and immune activities, transit-amplifying-like phenotype, and *TMEM106A* methylation. This subtype is deemed as the intermediate state of GINS2 and GINS5. Thus, further interventions should focus on how to convert GINS4 into GINS5 with better prognosis and sensitivity to immunotherapy, such as adoptive T-cell immunotherapy, cancer vaccine, and reprogramming the microenvironment (*Liu and Sun, 2021c*).

GINS5, an immune-activated subtype, is associated with serrated-like CRC, stronger immune activation, plentiful TMB and NAL, MSI, and CIMP-H, *BRAF* mutations, and favorable prognosis. This subtype commonly exhibits decent clinical outcomes due to the stronger immune activation. Spontaneously, GINS5 tumors respond well to immunotherapy. We also documented that tumors with high HDR score and mutations were predominantly assigned to GINS5, suggesting its nonnegligible potential to benefit from *PARP* inhibitors (*Liu et al., 2021a*). Indeed, four *PARP* inhibitors, including olaparib, niraparib, talazoparib, and KU-55933, were identified for more targeted or combined interventions for GINS5 tumors.

GINS6, a metabolic reprogramming subtype, is linked to accumulated fatty acids, enterocyte-like, and *BMP* activity. The lipid metabolisms are the most significant metabolic processes in GINS6. Also, the metabolomics results further demonstrated that GINS6 featured by abundant fatty acids metabolites, indicating GINS6 tumors could be intervened by metabolic inhibitors targeting fatty acid metabolisms. Interestingly, our pipeline determined four anticholesterol drugs containing atorvastatin, lovastatin, simvastatin, and fluvastatin, were specific for GINS6. Statins have been reported to attenuate cellular energy and outgrowth of cancers (*Ali et al., 2019*; *Beckwitt et al., 2018*), might provide extra options for this subtype.

From an interactome perspective, our study identified and diversely validated a high-resolution classification system, which could confidently serve as an ideal tool for optimizing decision-making for patients with CRC. The multifariously biological and clinical peculiarities of GINS taxonomy improve the understanding of CRC heterogeneity and facilitate clinical stratification and individuation management. Additionally, candidate specific-subtype agents provide more targeted or combined interventions for six subtypes, which also need to be validated in clinical settings. In this study, the GINS

taxonomy could be measured and reproduced using a simple PCR-based assay, making it attractive for clinical translation and implementation. Nevertheless, a prospective multicenter study is still imperative to further confirm the biological and clinical interpretability of six subtypes. To conclude, we believe this novel high-resolution taxonomy could facilitate more effective management of patients with CRC.

## Materials and methods

### Data source and specimen collection

A total of 6216 patients and 308 normal samples were enrolled from public databases. A merged discovery cohort consisted of 19 datasets (n=2167), another 19 independent datasets (n=3420) were used for validation, and 17 datasets including immunotherapy or radiotherapy annotations, cancer cell lines, patient-derived xenografts (PDX), and single-cell sequencing were applied for exploration. *Supplementary file 15* summarizes the data sources and details of this study. We also enrolled 214 clinical CRC samples from The First Affiliated Hospital of Zhengzhou University for further validation (*Supplementary file 3*).

### Construction of the gene interaction-perturbation network

Our gene interaction-perturbation pipeline applied the discovery cohort (n=2167) composed of 19 independent datasets from the same chip platform (Affymetrix Human Genome U133 Plus 2.0 Array, GPL570) as the tumor sample input and the GTEx cohort (n=308) as the normal sample input (*Figure 1* and *Supplementary file 15*). A pathway-derived analysis requires constructing the protein interaction functional networks projected by candidate genes (*Chen et al., 2021b*). The initial background network from the Reactome database (*Jassal et al., 2020*) included 6376 genes and 148,942 interactions, and fitted the biological scale-free network distribution in the discovery cohort ($R=-0.852$, $p<2.2e-16$; *Figure 1—figure supplement 1A*). The perturbation degree of gene interactions in the background network could measure the biological state of individual patients (*Chen et al., 2021a*). The global network perturbations are quantified via the interaction change of each gene pair, which is reasonably and effectively utilized to characterize the pathological condition at the individual level (*Chen et al., 2021b*; *Sahni et al., 2015*). In high-throughput profiles, we need to compute the relative perturbations of all gene pairs based on the benchmark vector. Since gene interactions are highly conservative within normal samples, we selected the average interactions of all normal samples as the benchmark vector. Gene interactions in each patient should be compared with the benchmark network, and the corresponding difference represents the gene interaction perturbation for that patient. Indeed, tumor samples displayed remarkably stronger perturbations relative to normal samples ($p<2.2e-16$; *Figure 1—figure supplement 1B*). The interaction perturbations of normal samples were much denser and less than tumor samples (*Figure 1—figure supplement 1C*). Collectively, 92.6% of all 148,942 gene pairs exhibited more dispersion in tumor samples than in normal samples by comparing the coefficient of variation (CV) of interaction perturbations ($P<2.2e-16$; *Figure 1—figure supplement 1D*). These results revealed that the interaction perturbations of normal samples were more stable, whereas a broader variation existed in tumor samples, making it possible to discover heterogeneous subtypes in CRC samples.

### Statistical analysis

The detailed methods and statistics were described in *Supplementary file 16*. All data processing, statistical analysis, and plotting were conducted in R 4.0.5 software. All statistical tests were two-sided. $p<0.05$ was regarded as statistically significant.

## Acknowledgements

This study was supported by The National Natural Science Foundation of China (81972663, 82173055, U2004112).

# Additional information

## Funding

| Funder | Grant reference number | Author |
|---|---|---|
| National Natural Science Foundation of China | 81972663 | Zhenqiang Sun |
| National Natural Science Foundation of China | 82173055 | Zhenqiang Sun |
| National Natural Science Foundation of China | U2004112 | Zhenqiang Sun |

The funders had no role in study design, data collection and interpretation, or the decision to submit the work for publication.

## Author contributions

Zaoqu Liu, Conceptualization, Data curation, Formal analysis, Validation, Investigation, Visualization, Methodology, Writing – original draft, Project administration, Resources, Software, Supervision, Writing – review and editing; Siyuan Weng, Qin Dang, Data curation, Writing – review and editing; Hui Xu, Validation, Writing – review and editing; Yuqing Ren, Chunguang Guo, Zhe Xing, Writing – review and editing; Zhenqiang Sun, Funding acquisition, Project administration; Xinwei Han, Supervision, Project administration

## Author ORCIDs

Zaoqu Liu ⓘ http://orcid.org/0000-0002-0452-742X
Xinwei Han ⓘ http://orcid.org/0000-0003-4407-4864

## Decision letter and Author response

Decision letter https://doi.org/10.7554/eLife.81114.sa1
Author response https://doi.org/10.7554/eLife.81114.sa2

# Additional files

## Supplementary files

• Supplementary file 1. The PAM-centroid distance classifier included 289 subtype-discriminant genes.

• Supplementary file 2. Clinical characteristics of six subtypes in validation datasets.

• Supplementary file 3. Details of baseline information in our in-house dataset.

• Supplementary file 4. The forward and reverse primers for qRT-PCR.

• Supplementary file 5. Clinical characteristics of six subtypes in our in-house dataset.

• Supplementary file 6. Publicly available gene signatures used in this study.

• Supplementary file 7. Sample set enrichment analysis delineated the biological attributes inherent to GINS subtypes. *P<0.05, **P<0.01, ***P<0.001.

• Supplementary file 8. Gene cluster 3 and 10 were identified by Mfuzz analysis.

• Supplementary file 9. The expression differences of 145 immunomodulators among six subtypes.

• Supplementary file 10. The proteome (Reverse Phase Protein Array) data available from the TCGA portal included only 26 immunomodulators.

• Supplementary file 11. GSEA results of 69 metabolic pathways from the Kyoto Encyclopedia of Genes and Genomes (KEGG) database.

• Supplementary file 12. Metabolite-protein interaction network (MPIN) was established via nine GINS6-specific genes with broad and tight connections with lipid metabolites.

• Supplementary file 13. Sample set enrichment analysis of ligand/receptor pairs expression in GINS subtypes. *NES and FDR are reported. NE, not enriched.*

• Supplementary file 14. The targeted pahtways and molecules of candidate durgs for six subtypes.

• Supplementary file 15. Details of data sources.

• Supplementary file 16. Detailed methods for this article.
• MDAR checklist

## Data availability

Public data used in this study are available in GEO, GTEx, TCGA, IMvigor210CoreBiologies, CCLE, GDSC, CTRP, and PRISM databases. Sequencing data available from GEO under accession codes GSE14333, GSE143985, GSE161158, GSE17537, GSE29621, GSE31595, GSE38832, GSE39084, GSE39582, GSE92921, GSE72970, GSE28702, GSE45404, GSE52735, GSE62080, GSE69657, GSE19860, GSE19862, GSE13067, GSE13294, GSE18088, GSE18105, GSE33113, GSE64256, GSE71222, GSE104645, GSE106584, GSE131418, GSE16125, GSE41258, GSE5851, GSE75315, GSE26682, GSE24551, GSE12945, GSE21510, GSE78220, GSE176307, GSE35144, GSE73255, GSE76402, GSE56699, and GSE81861. Normal tissue data is available from GTEx database (https://gtexportal.org). The TCGA-CRC multi-omics data, including RNA-seq (raw count), proteome (Reverse Phase Protein Array), HumanMethylation450 array, whole-exome sequencing (VarScan MAF files), and copy number variation (CNV) data, were derived from TCGA portal (https://portal.gdc.cancer.gov). Three datasets (n = 414) with immunotherapeutic annotations and expression profiles were derived from the following studies: Hugo and colleagues (*Hugo et al., 2016*) (GSE78220, n = 27), Rose and colleagues (*Rose et al., 2021*) (GSE176307, n = 89) and Mariathasan and colleagues (*Mariathasan et al., 2018*) (IMvigor210, n = 298). We retrieved 55 CRC cell lines with both transcriptome and metabolomics data (including 225 metabolites) from The Cancer Cell Line Encyclopedia (https://sites.broadinstitute.org/ccle, CCLE). Drug response and molecular data of human cancer cell lines were available from the Cancer Therapeutics Response Portal (CTRP, https://portals.broadinstitute.org/ctrp), Profiling Relative Inhibition Simultaneously in Mixtures (PRISM, https://depmap.org/portal/prism), and Genomics of Drug Sensitivity in Cancer (GDSC, https://www.cancerrxgene.org) datasets. Essential scripts to develop the GINS taxonomy have been uploaded to Github (https://github.com/Zaoqu-Liu/GINS; copy archived at swh:1:rev:de04c9140b621c687986834644bd9d318f9c440b).

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
