## [Editor Report]

In this paper, the investigators investigate CRC tumor heterogeneity by using a clustering method to understand perturbation of gene networks. The approach is robust and resulted in identification of significant differences in gene expression signals. The investigators identified six distinct gene interaction networks (GINs) that characterize tumor landscapes with varying degrees of oncogenic driver mutations, immune infiltration, and drug susceptibilities. These results provide a solid contribution to the field that, if validated, could be utilized as a useful predictive and prognostic correlative biomarkers in future clinical trials.

---

## [Decision Letter]

**Decision letter after peer review:**

Thank you for submitting your article "Gene interaction perturbation network deciphers a high-resolution taxonomy in colorectal cancer" for consideration by *eLife*. Your article has been reviewed by 3 peer reviewers, and the evaluation has been overseen by a Reviewing Editor and Wafik El-Deiry as the Senior Editor. The following individual involved in review of your submission has agreed to reveal their identity: Ruping Sun (Reviewer #1).

Essential revisions:

Overall the authors present a study evaluating colorectal cancer networks using unsupervised approach to assess gene interactions. The following reviews by experts in the field provide critical evaluation of the work as follows.

1) Address Major Concerns presented by all Reviewers in the Public Reviews, specifically from Reviewer 1.

2) All reviewers request clarifications or other added information to enhance clarify and accuracy of the manuscript. Please read these carefully and address as appropriate.

*Reviewer #1 (Recommendations for the authors):*

I found that much of the content of the first two paragraphs in the result section can be moved into methods. Authors can briefly mention the rationale and procedure of the method in the result section but should focus on the main findings of using such a method. In addition, the figure of the construction of the perturbation matrix (Figure S1) seems to be critical for readers to understand the method. I would suggest moving it into the main manuscript.

---

## [Author Response]

Reviewer #1 (Recommendations for the authors):I found that much of the content of the first two paragraphs in the result section can be moved into methods. Authors can briefly mention the rationale and procedure of the method in the result section but should focus on the main findings of using such a method. In addition, the figure of the construction of the perturbation matrix (Figure S1) seems to be critical for readers to understand the method. I would suggest moving it into the main manuscript.

Thanks for your thoughtful and constructive comments. After careful consideration, we totally agree with your opinion that the result section should focus on the main findings of using such a method. Thus, we've moved “Construction of the gene interaction-perturbation network” into the method parts (Red mark). In addition, as you suggested, we have moved the figure of the construction of the perturbation matrix to the main manuscript (Figure 1).